# Fair and Efficient Contribution Valuation for Vertical Federated Learning

**Zhenan Fan**[†*]**, Huang Fang**[†]**, Xinglu Wang**[‡*]**, Zirui Zhou**[*]**, Jian Pei**[‡]**, Michael P. Friedlander**[†]**, Yong Zhang**[*]

[†]University of British Columbia, {zhenanf,hgfang, mpf}@cs.ubc.ca
[‡]Simon Fraser University, {xinglu_wang_2,jpei}@cs.sfu.ca
[*]Huawei Technologies Canada, {zirui.zhou, yong.zhang3}@huawei.com

## Abstract

Federated learning is an emerging technology for training machine learning models across decentralized data sources without sharing data. Vertical federated learning, also known as feature-based federated learning, applies to scenarios where data sources have the same sample IDs but different feature sets. To ensure fairness among data owners, it is critical to objectively assess the contributions from different data sources and compensate the corresponding data owners accordingly. The Shapley value is a provably fair contribution valuation metric originating from cooperative game theory. However, its straight-forward computation requires extensively retraining a model on each potential combination of data sources, leading to prohibitively high communication and computation overheads due to multiple rounds of federated learning. To tackle this challenge, we propose a contribution valuation metric called vertical federated Shapley value (VerFedSV) based on the classic Shapley value. We show that VerFedSV not only satisfies many desirable properties of fairness but is also efficient to compute. Moreover, VerFedSV can be adapted to both synchronous and asynchronous vertical federated learning algorithms. Both theoretical analysis and extensive experimental results demonstrate the fairness, efficiency, adaptability, and effectiveness of VerFedSV.

## 1 Introduction

Creating powerful and robust machine learning models requires collecting enormous amounts of training data. However, in many industrial scenarios, training data is siloed across multiple companies, and data sharing is often impossible due to regulatory limits on data protection (Li et al., 2020). Federated learning (FL for short) is an emerging machine learning framework where a central server and multiple data owners, i.e., clients, collaboratively train a machine learning model without sharing their data (McMahan et al., 2017; Yang et al., 2019; Kairouz et al., 2019).

FL can further be classified into two main categories: horizontal federated learning (HFL) and vertical federated learning (VFL) (Yang et al., 2019). In HFL, various clients' data share the same feature space but have separate sample IDs. On the other hand, VFL assumes that the data owned by different clients share identical sample IDs but have distinctive features.

The success of FL depends on the active participation of motivated clients. Therefore it is vital to fairly evaluate the contributions from different clients as fair cooperation and reward can motivate active participation from players.

The Shapley value (Shapley, 1952) is a classical metric derived from cooperative game theory that is used to appropriately assess players' contributions. The Shapley value of a player is defined as the expected marginal contribution of the player over all possible subsets of the other players. The Shapley value is the only metric that satisfies all four requirements of Shapley's fairness criteria: *balance*, *symmetry*, *zero element*, and *additivity* (Dubey, 1975) (see Section B.1). Although the Shapley value has many desirable characteristics, its evaluation in the FL context requires repeatedly training and evaluating models learned on all possible subsets of clients. The corresponding communication and computation costs are exponential and often prohibitive in practice (Song et al., 2019; Wang et al., 2019; Fan et al., 2022).

Variants of the Shapley value make equitable data owner contribution assessment feasible in FL. Take its application in HFL as an example, Wang et al. (2020) proposed a contribution valuation

metric called the federated Shapley value (FedSV). The key idea is to calculate Shapley value for a client in each iteration of training and then aggregate these values over all iterations to determine the final contribution of the client. Its computation does not require model retraining and retains some, but not all, of Shapley's fairness criteria. Fan et al. (2022) enhanced the fairness of this method by considering all data owners, including those not selected in certain training iterations. This improvement is accomplished through low-rank matrix factorization techniques. While Fan et al. (2022) focus on HFL, this paper will serve as an extension towards VFL and complete the picture of contribution valuation in FL.

Compared to HFL, adapting the Shapley value to VFL faces another challenge because of the stronger model dependency in the vertical context. More precisely, the Shapley value computation requires us to form the model produced by all different subsets of clients. This requirement is easy to satisfy in HFL because the global model is defined as the additive aggregation of the local models, and thus we only need to aggregate local models from different subsets of clients (Wang et al., 2020). In VFL, however, the global model is the concatenation of local models that are not shared with the server. Thus, simply concatenating local models from different subsets of clients is not applicable.

In this paper, we concentrate on developing efficient and equitable methods for evaluating clients' contributions in VFL. We propose a contribution valuation metric called the *vertical federated Shapley value* (VerFedSV), where the clients' contributions are computed at multiple time stamps during the training process. We resolve the model concatenation problem by carefully utilizing clients' embeddings at different time stamps. We demonstrate that our design retains many desirable fairness properties and can be efficiently implemented without model retraining.

VFL algorithms can be divided into two categories: synchronous methods (Gong et al., 2016; Zhang et al., 2018; Liu et al., 2019), where periodic synchronizations among clients are required, and asynchronous ones (Hu et al., 2019; Gu et al., 2020; Chen et al., 2020), where clients are allowed to conduct local computation asynchronously. We show that VerFedSV applies to both synchronous and asynchronous VFL settings. Under the synchronous setting, we show that VerFedSV can be computed by leveraging some matrix-completion techniques. Although there are many similarities between synchronous and asynchronous VFL, contribution valuation in an asynchronous VFL setting is more complicated because the contribution of a client depends on not only the relevance of the client's data to the training task but also the client's local computational resource. We show that the design of VerFedSV can reflect the strength of clients' local computational resources under the asynchronous setting. To the best of our knowledge, we are the first to consider contributions w.r.t local computational resources.

Our contributions can be summarized as follows.

- We propose vertical federated Shapley value (VerFedSV) for VFL (Equation 1 and Definition 2), and show that VerFedSV satisfies many desirable properties of fairness (Theorem 1).
- Under the synchronous VFL setting, we show that VerFedSV can be computed by solving low-rank matrix completion problems for embedding matrices, which are proven to be approximately low-rank (Proposition 1). We also give an approximation guarantee of VerFedSV given the tolerance for matrix completion (Proposition 2).
- Under the asynchronous VFL setting, we show that VerFedSV can be directly computed and can reflect the strength of clients' local computational resources (Proposition 3).
- VerFedSV does not incur extra communication cost. Meanwhile, its computational cost can be further reduced by applying the Monte-Carlo sampling methods (Section D.1).

## 2   RELATED WORK

The Shapley value (Shapley, 1952) has broad applications (Gul, 1989). Dubey (1975) showed that Shapley value is the unique measure that satisfies the four fundamental requirements of fairness proposed by Shapley (1952). With a rich history and broad applications, it has been extended to contribution valuation in machine learning (ML) contexts (Ghorbani & Zou, 2019; Jia et al., 2019; Kwon et al., 2021; Sim et al., 2022). For example, Beta Shapley (Kwon & Zou, 2022) is a data valuation method tailored for ML applications, where the efficiency axiom can be relaxed and marginal contribution is affected by noisy data. Banzhaf value (Wang & Jia, 2023) is another data valuation method that, similarly, relaxes the efficiency axiom. It innovatively introduces the concept of a safety margin and addresses the randomness introduced by stochastic gradient descent. As privacy is an important aspect in ML, Wang et al. (2023) identify privacy leaks through the KNN-Shapley value score and propose DP-TKNN-Shapley, which offers a superior privacy-utility tradeoff. In the following, our discussion focuses on its application in FL.

The concept of the Shapley value was introduced into HFL to evaluate the contribution of federated participants (Song et al., 2019). The straight-forward computation of the Shapley value needs retraining the model an exponential number of times. To address this challenge, various methods have been proposed. Song et al. (2019) presented two gradient-based techniques to approximate the Shapley value. Wang et al. (2020) introduced the federated Shapley value, which can be determined from local model updates in each training iteration. The federated Shapley value does not need model retraining and preserves some but not all of the favorable qualities of the traditional Shapley value. Fan et al. (2022) enhanced the fairness of this method by considering all data owners, including those not selected in certain training iterations. This is achieved by leveraging low-rank matrix factorization techniques. Liu et al. (2022) proposed the Guided Truncation Gradient Shapley (GTG-Shapley) approach, which reconstructs FL models using gradient updates for Shapley value calculations, rather than retraining with varying combinations of FL participants, thereby enhancing both computational efficiency and approximation accuracy. Cosine Gradient Shapley Value (CGSV) (Xu et al., 2021) also avoids the need for model retraining and validation processes by exploiting information available during training, that is, the alignment between the model updates uploaded by an individual agent and the aggregated model derived from all participating agents. Meanwhile, Xu et al. (2021) propose an incentive mechanism that rewards agents with higher contributions through a higher-quality model. However, this realization of incentives may lead to non-convergent behavior in the model. To address this, Lin et al. (2023) further explore the trade-off between asymptotic performance and fairness. Our method shares a similar idea in the sense of leveraging training-time information, but the difference is that we cater to synchronous and asynchronous VFL scenarios. Beyond the computational challenge, Yang et al. (2022) find that the Shapley value only considers the marginal contribution on model performance and propose to enhance the Shapley value with other influencing factors, such as data quality, level of cooperation, and risk factor, to assess contributions. Apart from its use in contribution valuation, the Shapley value has inspired an adaptive weighting mechanism (Sun et al., 2023) to enhance the robustness of FL.

Wang et al. (2019) and more recently Han et al. (2021) extended the notion of data Shapley value to VFL. As discussed before, the need for retraining the model for different subsets of clients is a bottleneck of Shapley value computation in federated learning. Wang et al. (2019); Han et al. (2021) solved this problem by introducing model-independent utility functions, where the contribution of a client does not depend on the performance of the final model, and thus does not require retraining of the model. In particular, Wang et al. (2019) suggested using the situational importance (SI) (Achen, 1982), which computes the difference between the embeddings with true features and expected features. However, computing SI requires knowing the expectation of each feature and can be impractical under VFL. Han et al. (2021) suggested to use the conditional mutual information (CMI) (Brown et al., 2012), which computes the tightness between the label and features. However, computing CMI requires every client to access the labels, which may also be impractical under VFL and leak privacy. Besides the above shortcomings, the model-independent utility function itself may cause some fairness issues when the VFL is conducted asynchronously. Specifically, under the asynchronous setting, the contribution of a client not only is related to the quality of the local dataset but also depends on the power of local computational resources. Model-independent utility functions cannot fully reflect a client's contribution in this scenario. (See Proposition 3 and Section 7.2 for details.) In this work, we instead use a model-dependent utility function and resolve the requirement of retraining the model by periodically evaluating the contribution metric during the training process.

## 3  VERTICAL FEDERATED LEARNING

In this section, we review the standard VFL framework. In this framework, $M$ clients and one server collaborate to train a machine learning model on $N$ data samples $\{(x_i \in \mathbb{R}^d, y_i \in \{\pm 1\})\}_{i=1}^N$, where $x_i$ is the $i$-th feature vector and $y_i$ is its associated label. We assume that the index $i$ is globally shared between the server and the $M$ clients. This assumption is reasonable, as data alignment methods in VFL protocols (Cheng et al., 2019) are routinely employed to ensure synchronization across clients. This method ensures that only samples shared by all clients are retained for VFL, each identified by a unique sample ID. For simplicity, we posit that these sample IDs are the same as the global sample indices $[N] = \{1, \cdots, N\}$. Here, the notation $[\cdot]$ means the set of consecutive integers.

Under the VFL framework, every feature vector $x_i$ is distributed across $M$ clients. Represent the portion of the feature vector held by client $m$ as $x_i^{(m)} \in \mathbb{R}^{d^{(m)}}$, where $d^{(m)}$ specifies the feature dimension associated with client $m$. In other words, if we collect the feature vectors from all clients and concatenate them, the $x_i$ will be reconstructed, *i.e.*, $x_i = [x_i^1, \ldots, x_i^M]$ and $d = \sum_{m=1}^M d^{(m)}$. Here, $[\cdot, \ldots, \cdot]$ is the concatenation operation.

The local data set for client $m$ is $\mathcal{D}^{(m)} = \left\{ x_i^{(m)} : i \in [N] \right\}$. The server maintains all the labels $\mathcal{D}^s = \{ y_i : i \in [N] \}$. The collaborative training problem can be formulated as $\min_{\theta_1, \ldots, \theta_M} \frac{1}{N} \sum_{i=1}^{N} \ell \left( \theta_1, \ldots, \theta_M; \{x_i, y_i\} \right)$, where $\theta_m \in \mathbb{R}^{d^{(m)}}$ denotes the training parameters of the $m$-th client and $\ell(\cdot)$ denotes the loss function. For a wide range of models, such as linear and logistic regression, and support vector machines, the loss function has the form $\ell \left( \theta_1, \ldots, \theta_M; \{x_i, y_i\} \right) := f \left( h_i; y_i \right)$, where $h_i = \sum_{m=1}^{M} h_i^{(m)}$, $h_i^{(m)} = \left\langle \theta_m, x_i^{(m)} \right\rangle$, and $f(\cdot; y)$ is a differentiable function for any $y$. For each client $m$, the term $h_i^{(m)}$ can be viewed as the client's embedding of the local data point $x_i^{(m)}$ and the local model $\theta_m$. To preserve privacy, clients are not allowed to share their local data set $\mathcal{D}^{(m)}$ or local model $\theta^{(m)}$ with other clients or with the server. Instead, clients can share only their local embeddings $\left\{ h_i^{(m)} : i \in [N] \right\}$ to the server for training.

For simplicity, here we consider a binary classification problem. All of our theoretical results can be easily extended to a multi-class classification problem with $l > 2$ classes, where $\theta_m \in \mathbb{R}^{d^{(m)} \times l}$ and $h_i^{(m)} = \theta_m^\mathsf{T} x_i^{(m)} \in \mathbb{R}^l$. Moreover, in practice, our method also works for more general local embedding functions, i.e., $h_i^{(m)} = g \left( \theta_m; x_i^{(m)} \right)$, where $g$ can be a neural network with weights $\theta_m$. We empirically verify this in Section 7.

Asynchronous VFL is favorable when the strength of computational resources varies greatly across clients. This strength concerns two aspects: 1). The batch size of embeddings that a client can process influenced by its hardware (e.g., CPU/GPU) capability. 2). The upload speed influenced by its transmission power. It is reasonable to assume that (Dinh et al., 2020) download times are negligible compared to upload times, given that downlinks tend to have higher bandwidths, and the server's transmission power far exceeds that of the client.

Formally, we define the strength of clients' local computational resources that reflect the above-mentioned two factors as follows.

**Definition 1.** *Let $\Delta t > 0$ be the unit time for one iteration including the time for communication. The strength of client $m$'s local computational resource is represented by a positive integer $\tau_m \in [1, N]$, such that client $m$ can compute and upload at most $\tau_m$ local embeddings during each time interval $\Delta t$.*

## 4 VERTICAL FEDERATED SHAPLEY VALUE

In the following, we briefly introduce why Shapley value is a fair valuation metric. The Shapley value is built upon a black-box utility function. Under the FL setting, this utility function outputs the performance of the collaboratively trained model. Since each evaluation of utility involves retraining a model, we denote this utility function as $U^{\text{retrain}} : 2^{[M]} \to \mathbb{R}$. It provides a utility score for the model trained by any client subset $S \subseteq [M]$. Given utility function $U^{\text{retrain}}$, the Shapley value is said to be fair because it satisfies four fundamental requirements: Symmetry, zero element, additivity, and balance (See Section B.1 for details). It has been shown that the only valuation metric satisfying four requirements is the Shapley value (Dubey, 1975; Ghorbani & Zou, 2019). Formally, the Shapley value assigned to client $m$, denoted by $s_m$, is computed by $s_m = \frac{1}{M} \sum_{S \subseteq [M] \setminus \{m\}} \frac{1}{\binom{M-1}{|S|}} \left[ U^{\text{retrain}}(S \cup \{m\}) - U^{\text{retrain}}(S) \right]$.

However, computing $s_m$ is impractical in the FL context because the evaluation of the utility function $U^{\text{retrain}}$ requires extensive model retraining (Ghorbani & Zou, 2019; Wang et al., 2020). To it computationally practical in FL context, as discussed before, Wang et al. (2020) recently proposed the federated Shapley value (FedSV) for HFL, which computes the Shapley values for clients periodically during the training and then reports the summation over all the periods as the final results. We extend this idea to the VFL context and define a new utility function. Suppose we pre-determine $T$ time stamps for contribution valuation. Denote by $[T] = \{1, \ldots, T\}$ the set of all time stamps. At each time $t \in [T]$, we define the utility function $U_t : 2^{[M]} \to \mathbb{R}$ such that for any subset of clients $S \subseteq [M]$, the utility $U_t(S)$ denotes the decrease in loss made by the clients in $S$ during the time

period $[t-1, t]$, i.e.,

$$U_t(S) = \frac{1}{N} \sum_{i=1}^{N} f\left(\sum_{m=1}^{M} \left(h_i^{(m)}\right)^{(t-1)}; y_i\right) - \frac{1}{N} \sum_{i=1}^{N} f\left(\sum_{m \in S} \left(h_i^{(m)}\right)^{(t)} + \sum_{m \notin S} \left(h_i^{(m)}\right)^{(t-1)}; y_i\right),$$

(1)

where $\left(h_i^{(m)}\right)^{(t)}$ is the local embedding of data point $x_i^{(m)}$ by client $m$ at time $t$. The idea behind this definition is that if client $m$ participates in the training during the period $[t-1, t]$, then we use the client's latest embedding $\left(h_i^{(m)}\right)^{(t)}$ for evaluation; otherwise, we use the previous embedding $\left(h_i^{(m)}\right)^{(t-1)}$. Then, we formally define the vertical federated Shapley value (VerFedSV).

**Definition 2.** *Given $T$ predetermined contribution valuation time stamps, the VerFedSV for any client $m \in [M]$ is $s_m = \frac{1}{MT} \sum_{t=1}^{T} \sum_{S \subseteq [M] \setminus \{m\}} \frac{1}{\binom{M-1}{|S|}} [U_t(S \cup \{m\}) - U_t(S)]$.*

Note that computing VerFedSV does not require retraining the model due to Equation 1. Next, theorem 1 justifies the fairness of VerFedSV. Please refer to Section B.1 for proof.

**Theorem 1** (Fairness Guarantee). *Define $U : 2^{[M]} \to \mathbb{R}$ as the averaged utility function spanning the entire training process, formulated as $U(S) = \frac{1}{T} \sum_{t=1}^{T} U_t(S)$. The VerFedSV (Definition 2) satisfies four requirements on fairness (See Section B.1 for details).*

## 5 COMPUTATION OF VERFEDSV UNDER SYNCHRONOUS SETTING

Almost all the synchronous VFL algorithms (Gong et al., 2016; Zhang et al., 2018; Liu et al., 2019) share the same framework: in each iteration, every client uploads local embeddings for the same batch of data points to the server, downloads gradient information from the server, and then conducts local updates. The main difference among those algorithms is the scheme of local updates. For clarity, we consider the vanilla stochastic gradient descent algorithm for VFL, i.e., FedSGD (Liu et al., 2019). It is worth mentioning that VerFedSV computation solely relies on the clients' embeddings and is independent of the local update rule of the underlying VFL algorithm. Consequently, VerFedSV can be easily extended to other synchronous VFL algorithms.

Refer to Section A.1 for details on one training iteration in FeDSGD. Here, we describe how batch size is determined. The time interval between training iterations is first negotiated and determined. Then, Definition 1 gives the maximum number of local embeddings $\tau_m$ that can be processed by client $m$ within a single iteration. The server collects information from clients and calculates $\tau := \min\{\tau_m \mid m \in [M]\}$. The batch size is set to be no larger than $\tau$. In the following, we denote the mini-batch of $t$-th iteration as $B^{(t)}$, and the embedding of $i$-th sample for $m$-th clients as $\left(h_i^{(m)}\right)^{(t)}$.

Now we show how to compute VerFedSV with the FedSGD algorithm. Under the synchronous setting, we set the time stamps for contribution valuation to be the ends of training iterations. The key challenge in computing VerFedSV is that, in each iteration, we only have access to the local embeddings for the current mini-batch $B^{(t)}$, i.e., $H^{(t)} = \left\{ \left(h_i^{(m)}\right)^{(t)} : m \in [M], \ i \in B^{(t)} \right\}$. However, computing VerFedSV (Definition 2) requires *all* the local embeddings in each iteration, i.e., $\hat{H}^{(t)} = \left\{ \left(h_i^{(m)}\right)^{(t)} : m \in [M], \ i \in [N] \right\}$. Therefore, we want to obtain reasonable approximations for the missing local embeddings.

**Embedding matrix** For each client $m$, we define the embedding matrix $\mathcal{H}^{(m)} \in \mathbb{R}^{T \times N}$ where each element $(t, i)$ is defined as $\mathcal{H}_{t,i}^{(m)} = \left(h_i^{(m)}\right)^{(t)}$. Due to the mini-batch setting, we can only make partial observations $\{\mathcal{H}_{t,i}^{(m)} : t \in [T], \ i \in B^{(t)}\}$. We notice that the embedding matrices can be decomposed as $\mathcal{H}^{(m)} = \Theta_m X_m$ where $\Theta_m = \begin{bmatrix} \theta_m^{(1)} & \cdots & \theta_m^{(T)} \end{bmatrix}^{\mathsf{T}}$ and $X_m = \begin{bmatrix} x_1^{(m)} & \cdots & x_N^{(m)} \end{bmatrix}$. Note that when $d^{(m)} < \min\{T, N\}$, the embedding matrix $\mathcal{H}^{(m)}$ is low-rank because $\mathrm{rank}(\mathcal{H}^{(m)}) \leq \min\{\mathrm{rank}(\Theta_m), \mathrm{rank}(X_m)\} \leq d^{(m)}$. When $d^{(m)} \geq \min\{T, N\}$, we observe that the model matrix $\Theta_m$ is approximately low-rank due to the similarity of local models between successive training

iterations. The data matrix $X_m$ can also be approximately low-rank due to the similarity between local data points. Our following proposition theoretically formalizes this observation. Before stating the result, we first give a formal definition of approximately low-rankness, proposed by Udell & Townsend (2019).

**Definition 3** ($\epsilon$-rank, (Udell & Townsend, 2019, Def. 2.1)). *Let $X \in \mathbb{R}^{m \times n}$ be a matrix and $\epsilon > 0$ a tolerance. The $\epsilon$-rank of $X$ is defined as $\mathrm{rank}_\epsilon(X) := \min\{\mathrm{rank}(Z) : Z \in \mathbb{R}^{m \times n}, \|Z - X\|_{\max} \le \epsilon\}$, where $\|\cdot\|_{\max}$ is the absolute maximum matrix entry. Consequently, $k = \mathrm{rank}_\epsilon(X)$ signifies the minimal integer such that $X$ can be approximated by a rank-$k$ matrix within an $\epsilon$-tolerance.*

The following proposition characterizes the approximate rank of the embedding matrices $\mathcal{H}^{(m)}$.

**Proposition 1** ($\epsilon$-rank of the embedding matrix). *Assume that the function $f(\cdot; y)$ is $L$-smooth for any label $y$, the local data sets are normalized, i.e., $\|x_i^{(m)}\| = 1$ for all $m \in [M]$ and $i \in [N]$, and the learning rate is defined as $\eta^{(t)} := 1/t$. Then for any $\epsilon > 0$ and any $m \in [M]$, $\mathrm{rank}_\epsilon(\mathcal{H}^{(m)}) \le \min\left\{ d^{(m)}, \left\lceil \frac{L \log(T)}{\epsilon} \right\rceil, \mathcal{N}\left( \mathcal{D}^{(m)}, \frac{\epsilon}{\gamma^{(m)}} \right) \right\}$, where $\mathcal{N}(\cdot, \cdot)$ is the covering number, and $T$ is the number of total iterations.*

**Low-rank matrix completion** Proposition 1 shows that the embedding matrix is approximately low-rank. For any client $m \in [M]$, we formulate the following factorization-based low-rank matrix completion problem to complete the embedding matrix $\mathcal{H}^{(m)}$:

$$\underset{\substack{W^{(m)} \in \mathbb{R}^{T \times r} \\ H^{(m)} \in \mathbb{R}^{N \times r}}}{\mathrm{minimize}} \quad \sum_{t=1}^{T} \sum_{i \in B^{(t)}} \left( \mathcal{H}_{t,i}^{(m)} - \left( w_t^{(m)} \right)^\intercal h_i^{(m)} \right)^2 + \lambda(\|W^{(m)}\|_F^2 + \|H^{(m)}\|_F^2), \tag{2}$$

where $r$ is a user-specified rank parameter, $\lambda$ is a positive regularization parameter, $\|\cdot\|_F$ is the Frobenius norm, and $w_t^{(m)}$ and $h_i^{(m)}$, respectively, are the $t$-th and the $i$-th row vectors of the matrices $W^{(m)}$ and $H^{(m)}$. The rank parameter $r$ can be determined via Proposition 1.

The low-rank matrix completion model (Equation 2) was first used in completing the information for recommender systems (Koren et al., 2009). Its effectiveness has been extensively studied both theoretically and empirically (Keshavan, 2012; Sun & Luo, 2016). We can therefore adopt well-established matrix-completion methods (Yu et al., 2014; Chin et al., 2016) for solving the problem in Equation 2.

Note that the matrix completion problem (Equation 2) can be done on the client side in parallel, i.e., clients will independently complete their embedding matrices. It is known that if we solve the matrix completion problem in Equation 2 using the proximal gradient method, then the computational complexity for each iteration is $\Omega(r(T + N + \tau TN))$ (Udell et al., 2016). Here, the number of steps $\tau$ required by the proximal gradient method is typically considered to be a small constant. We numerically show the time required for solving the matrix completion problem in Equation 2 in Section D.1.

**Approximation guarantee** The following proposition shows that if we can obtain a factorization with a small error, i.e., $\mathcal{H}^{(m)} \approx W^{(m)}(H^{(m)})^\intercal$ via solving Equation 2, then we can also guarantee a good approximation to the true VerFedSV.

**Proposition 2** (Approximation guarantee). *Define the factorization error as $\epsilon := \frac{1}{M} \sum_{m=1}^{M} \|\mathcal{H}^{(m)} - W^{(m)}(H^{(m)})^\intercal\|_{\max}$. For any client $m \in [M]$, let $\hat{s}_m$ denote the VerFedSV computed with $\{W^{(m)}, H^{(m)} : m \in [M]\}$, i.e., $\hat{s}_m = \frac{1}{MT} \sum_{t=1}^{T} \sum_{S \subseteq [M] \setminus \{m\}} \frac{1}{\binom{M-1}{|S|}} [\hat{U}_t(S \cup \{m\}) - \hat{U}_t(S)]$, where*

$$\hat{U}_t(S) = \frac{1}{N} \sum_{i=1}^{N} f\left( \sum_{m=1}^{M} (w_{t-1}^{(m)})^\intercal h_i^{(m)}; y_i \right) - \frac{1}{N} \sum_{i=1}^{N} f\left( \sum_{m \in S} (w_t^{(m)})^\intercal h_i^{(m)} + \sum_{m \notin S} (w_{t-1}^{(m)})^\intercal h_i^{(m)}; y_i \right).$$

*If the function $f(\cdot; y)$ is $G$-Lipschitz for any label $y$, then $|\hat{s}_m - s_m| \le 2G\epsilon$ for all client $m \in [M]$.*

## 6 COMPUTATION OF VERFEDSV UNDER ASYNCHRONOUS SETTING

Algorithms using synchronous computation are inefficient when applied to real-world VFL tasks, especially when clients' computational resources are unbalanced. In this section, we show how to equip VerFedSV with asynchronous VFL algorithms as follows.

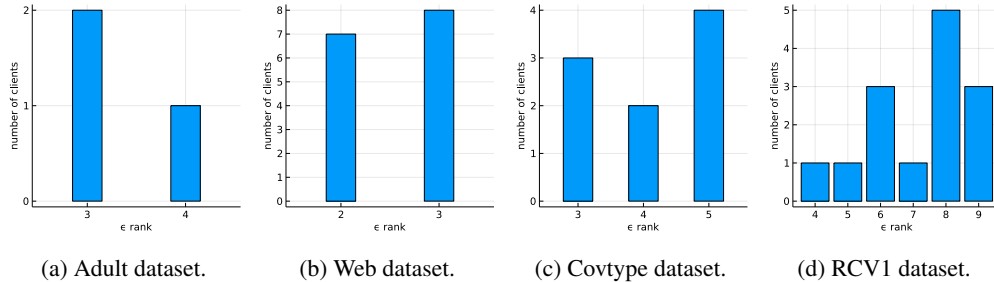

(a) Adult dataset.   (b) Web dataset.   (c) Covtype dataset.   (d) RCV1 dataset.

Figure 1: Approximated $\epsilon$-rank of embedding matrices.

We briefly introduce the vertical asynchronous federated learning (VAFL) algorithm (Chen et al., 2020). This algorithm allows each client to run stochastic gradient algorithms without coordination with other clients. The server maintains the latest embeddings for each sample. At any time, the clients can update an embedding or query gradient from the server. Please refer to Section A.2 for the detailed training process of the VAFL algorithm.

Next, we show how to compute VerFedSV with the VAFL algorithm. Under the asynchronous setting, there is no definition of training iterations from the perspective of the server. According to Definition 2, we can pre-determine $T$ time stamps for contribution valuation. At any contribution valuation time $t \in [T]$, the server keeps the set of embeddings $H^{(t-1)}$ from $t-1$ and $H^{(t)}$ from $t$. Here, $H^{(t)} = \{\left(h_i^{(m)}\right)^{(t)} \mid m \in [M], \ i \in [N]\}$. At $t = 0$, we initialize the server's embeddings with all zero, i.e., $\left(h_i^{(m)}\right)^{(0)} = 0, \ \forall m \in [M], \ \forall i \in [N]$. With these embeddings, VerFedSV is computed according to Equation 1 and Definition 2.

A careful reader may ask why there is no need to use matrix completion under the asynchronous setting. A short answer is that, under this setting, the contribution of a client is related to both the quality of the local dataset and the power of local computational resources, which is reflected by the parameter $\tau_m$ (Definition 1). More precisely, at any contribution valuation time point $t \in [T]$, for any client $m \in [M]$, there exists $B \subset [N]$ such that only the embeddings corresponding to $B$ are updated, i.e., $\left(h_i^{(m)}\right)^{(t)} \neq \left(h_i^{(m)}\right)^{(t-1)}, \forall i \in B$, and $\left(h_i^{(m)}\right)^{(t)} = \left(h_i^{(m)}\right)^{(t-1)}, \forall i \notin B$, where the size of $B$ is proportional to $\tau_m$. Then according to Equation 1, for any $S \subset [M] \setminus \{m\}$, we have $U_t(S \cup m) - U_t(S) = \frac{1}{N} \sum_{i \in B} \left[ f\left( \left(h_i^{(m)}\right)^{(t-1)} + (h_i^{(-m)})^{(t)}; y_i \right) - f\left( \left(h_i^{(m)}\right)^{(t)} + (h_i^{(-m)})^{(t)}; y_i \right) \right]$, where $(h_i^{(-m)})^{(t)} := \sum_{k \in S} (h_i^k)^{(t)} + \sum_{k \notin S \cup \{m\}} (h_i^k)^{(t-1)}$.

Therefore, we can see that the contribution of client $m$ is proportional to $\tau_m$, which is indeed an important feature of asynchronous VFL algorithms. Thus, if we conduct the embedding matrix completion under the synchronous setting, then we will lose this important feature, which is unfair to the clients with more powerful local computational resources.

The above discussion also suggests that VerFedSV can motivate clients to communicate more with the server in the asynchronous setting, i.e., dedicating more of their local computational resources. The following proposition demonstrates that two clients with the same local datasets but different communication frequencies receive different valuations. More specifically, the one who communicates more with the server receives a higher valuation.

**Proposition 3** (More work leads to more rewards). *Consider a simple case: we have clients 1 and 2 with identical local datasets where $x_i^1 = x_i^2 = x_i$ for all $i \in [N]$. However, their communication frequencies differ: while client 1 consistently sends local embeddings to the server, client 2 only does so with a probability of $\rho \in [0, 1]$ each time client 1 communicates.*

*Suppose that the loss function $g(\theta) = \frac{1}{N} \sum_{i=1}^{N} f(\langle \theta, x_i \rangle, y_i)$ is $\mu$-strongly convex for some $\mu > 0$ and $\theta^*$ is the global minimum point, $\theta_1$ and $\theta_2$ are initialized at 0, and we stop training when we reach the optimum point. Then it follows that $\mathbb{E}[\theta_1^*] = \frac{1}{1+\rho} \theta^*$ and $\mathbb{E}[\theta_2^*] = \frac{\rho}{1+\rho} \theta^*$.*

*Furthermore, if we only do one time of contribution valuation when the training ends, $\mathbb{E}[s_1] \geq \mathbb{E}[s_2] + \mu \left(\frac{1-\rho}{1+\rho}\right)^2 \|\theta^*\|^2$, where $s_1$ and $s_2$ are the VerFedSV for clients 1 and 2, respectively.*

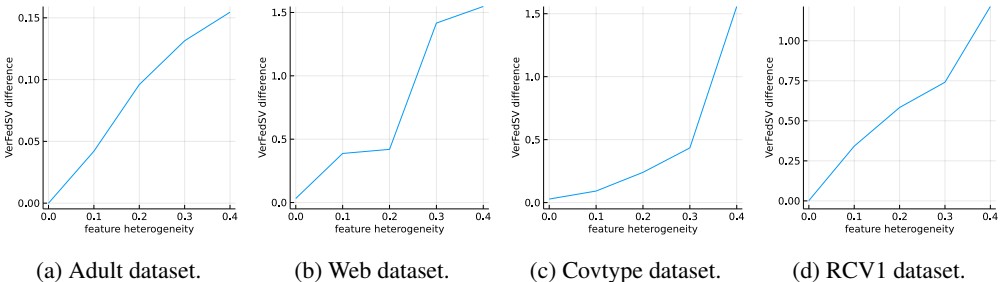

Figure 2: Relative VerFedSV difference v.s. feature heterogeneity.

## 7 EXPERIMENTS

We conduct extensive experiments on real-world datasets, including *Adult* (Zeng et al., 2008), *Web* (Platt, 1998), *Covtype* (Blackard & Dean, 1999), and *RCV1* (Lewis et al., 2004). The detailed setup is elaborated in Section C. Our code is submitted in the supplementary material.

### 7.1 SYNCHRONOUS VERTICAL FEDERATED LEARNING

In this section, we first empirically verify that the embedding matrices are indeed approximately low-rank (Proposition 1). Next, we demonstrate that VerFedSV satisfies the fairness property under synchronous setting (Theorem 1). More precisely, we verify that clients with similar features should get similar valuations and clients with randomly generated features should get low valuations.

**Rank of the embedding matrix** It is required to access the full embedding matrix in order to verify its low-rankness. Thus, we set the batch size equal to the number of data points for all clients. Considering the experiment doesn't focus on scalability w.r.t the total numbers of clients $M$, we set $M$ to a moderate value. For the *Adult*, *Web*, *Covtype* and *RCV1* datasets, respectively, $M = 3, 15, 9, 14$. Since computing the $\epsilon$-rank (Definition 3) is NP-hard (Udell & Townsend, 2019), we instead compute the rank of its truncated singular value decomposition as an approximation. More precisely, given an embedding matrix $\mathcal{H}^{(m)} \in \mathbb{R}^{T \times N}$ for client $m$, let $\sigma_1 \geq \sigma_2 \geq \ldots \geq \sigma_p$ be its ordered singular values, where $p = \min\{T, N\}$. Then given $\epsilon \in (0, 1)$, we define its approximated $\epsilon$-rank as $\widehat{\text{rank}}_\epsilon(\mathcal{H}^{(m)}) = \max\{r \in [1, p] \mid \sigma_r \geq \epsilon \cdot \sigma_1\}$.

There will be $M$ rank values for $M$ clients' embedding matrices. The histogram of these approximated $\epsilon$-rank values is shown in Figure 1, where $\epsilon = 10^{-3}$, the x-axis represents the approximated $\epsilon$-rank and the y-axis represents the number of clients with associating x-axis value of approximated $\epsilon$-rank. The plot shows that the embedding matrices are low-rank on all the datasets.

**VerFedSV for similar features** Besides the original clients, for each data set, we add 5 more clients whose features are identical to client 1 but with different levels of perturbations. More precisely, for new client $i \in \{1, \ldots, 5\}$, we add white Gaussian noise to $(i-1)10\%$ percent of the local features, which we denote as the feature heterogeneity. Then we measure the relative difference between the original client 1 and the new clients, i.e., for any new client $i \in \{1, \ldots, 5\}$, let $\text{diff}_i := \frac{|s - s_i|}{s}$, where $s$ is the VerFedSV for the original client 1 and $s_i$ is the VerFedSV for the new client $i$. We show a plot of relative VerFedSV difference v.s. feature heterogeneity in Figure 2, where the numbers of clients for the *Adult*, *Web*, *Covtype* and *RCV1* datasets are, respectively, $M = 8, 20, 14, 19$. We can see that the relative VerFedSV difference is proportional to the feature heterogeneity. Besides, when the feature heterogeneity is equal to 0, i.e., two clients have identical features, the relative VerFedSV difference is exactly 0 for *Adult* dataset, and is nearly 0 for the *Web*, *Covtype* and *RCV1* datasets, where the inexactness is due to the Monte-Carlo sampling.

**VerFedSV for random features** Besides the original clients, for each data set, we add 5 more clients whose features are randomly generated according to different distributions. Specifically, for the new client $i \in \{1, \ldots, 5\}$, the features are generated from Gaussian distribution with a mean equal to $i$ and variance equal to $i^2$. Table 1 shows the percentage of each client's VerFedSVs relative to the sum of all VerFedSVs, where the numbers of clients for the *Adult*, *Web*, *Covtype* and *RCV1* datasets are, respectively, $M = 8, 20, 14, 19$. As shown in table, regardless of the distributions, clients with randomly generated features receive much lower valuations than the regular clients for all the datasets.

Table 1: Percentage of clients' VerFedSVs in the total sum of VerFedSVs.

|  | Synchronous setting | | | | Asynchronous setting | | | |
|---|---|---|---|---|---|---|---|---|
|  | Adult | Web | Covtype | RCV1 | Adult | Web | Covtype | RCV1 |
| all regular clients | 99.88 | 99.73 | 97.77 | 93.84 | 97.91 | 98.39 | 98.57 | 91.90 |
| artificial client 1 | 0.01 | 0.02 | 0.39 | 1.86 | 0.93 | 0.26 | 0.35 | 1.66 |
| artificial client 2 | 0.05 | 0.06 | 0.57 | 0.43 | 0.06 | 0.31 | 0.21 | 1.75 |
| artificial client 3 | 0.03 | 0.06 | 0.91 | 2.37 | 0.33 | 0.36 | 0.22 | 1.64 |
| artificial client 4 | 0.01 | 0.07 | 0.36 | 0.31 | 0.43 | 0.40 | 0.25 | 1.50 |
| artificial client 5 | 0.02 | 0.06 | 0.01 | 1.18 | 0.35 | 0.28 | 0.39 | 1.54 |

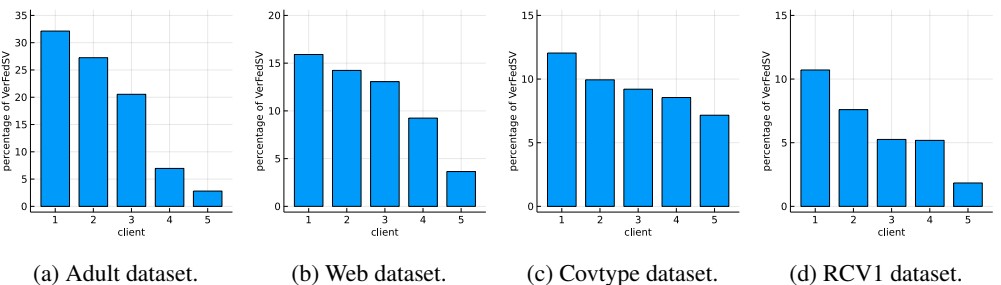

(a) Adult dataset.  (b) Web dataset.  (c) Covtype dataset.  (d) RCV1 dataset.

Figure 3: VerFedSVs for clients with different communication frequencies.

## 7.2 ASYNCHRONOUS VERTICAL FEDERATED LEARNING

In this section, we show that VerFedSV not only satisfies the fairness property under asynchronous setting (Theorem 1), but can also reflect how frequently clients report (Proposition 3). Specifically, during the training, we let clients communicate with the server at different frequencies. For all the datasets, we asynchronously train the model for 20 seconds and do the valuation every 0.04 second, i.e., there are $T = 500$ contribution valuation time points (Definition 2).

**Impact of communication frequency** Besides the original clients, for each data set, we add 5 more clients whose features are identical to client 1 but with different levels of communication frequencies. More precisely, the new client $i \in \{1, \ldots, 5\}$ communicates with the server every $0.01i$ second, i.e., the new client 1 has the highest communication frequency, and the new client 5 has the lowest. We show a plot in Figure 3 of the percentage of the new clients' VerFedSVs in the total sum of VerFedSVs, where the numbers of clients for the *Adult*, *Web*, *Covtype* and *RCV1* datasets are, respectively, $M = 8, 20, 14, 19$. We can see that the percentage of VerFedSV is proportional to the communication frequency.

**VerFedSV for random features** Besides the regular clients, for each data set, we add 5 more clients whose features are randomly generated according to standard Gaussian distribution. They have communication frequencies such that the new client $i \in \{1, \ldots, 5\}$ communicates with the server every $0.01i$ seconds. We show in Table 1 the percentage of the clients' VerFedSVs in the total sum of VerFedSVs, where the numbers of clients for the *Adult*, *Web*, *Covtype* and *RCV1* datasets are, respectively, $M = 8, 20, 14, 19$. Regardless of the communication frequencies, the clients with randomly generated features receive much lower valuations than the regular clients for all the datasets.

## 8 CONCLUSION AND INSIGHTS

In this paper, we propose a contribution valuation metric, i.e., vertical federated Shapley value (VerFedSV). We demonstrate theoretically and empirically that VerFedSV satisfies desirable properties for fairness and is adaptable to both synchronous and asynchronous VFL settings. There are a few interesting future directions. We notice that when we keep adding clients with identical features, the total of the VerFedSV increases. This suggests that some clients may "cheat" by constructing new clients with identical features to receive unjustifiable rewards in the end. One potential solution is to implement secure statistical testing before FL training and exclude clients with extremely similar datasets. It is also interesting to explore whether VerFedSV can be integrated into differentially private VFL algorithms.

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

# A    DETAILS ON ADOPTED FEDERATED LEARNING ALGORITHM

## A.1    SYNCHRONOUS SETTING

We adopt the FedSGD algorithm (Liu et al., 2019) for synchronous federated learning. In the following, we show the sketch of one training iteration of the FedSGD. In each iteration $t$, the FedSGD algorithm executes the following steps:

1. The server selects a mini-batch $B^{(t)} \subset [N]$ containing the global indices of samples;

2. Each client $m \in [M]$ computes local embeddings $\{\left(h_i^{(m)}\right)^{(t)} = \langle x_i^{(m)}, \theta_m^{(t)} \rangle \mid i \in B^{(t)}\}$, where $\theta_m^{(t)}$ is the client's current model and sends them to the server;

3. The server computes gradient information $\left\{ g_i^{(t)} := \frac{\partial f(h_i^{(t)}; y_i)}{\partial h_i^{(t)}} \;\middle|\; i \in B^{(t)},\ h_i^{(t)} = \sum_{m=1}^M \left(h_i^{(m)}\right)^{(t)} \right\}$ and sends it to every client $m \in [M]$;

4. Each client $m \in [M]$ updates the local model via $\theta_m^{(t+1)} \leftarrow \theta_m^{(t)} - \frac{\eta^{(t)}}{|B^{(t)}|} \sum_{i \in B^{(t)}} g_i^{(t)} x_i^{(m)}$, where $\eta^{(t)}$ is the learning rate at the $t$-th iteration.

## A.2    ASYNCHRONOUS SETTING

We follow the vertical asynchronous federated learning (VAFL) algorithm proposed by Chen et al. (2020), where the algorithm allows each client to run stochastic gradient algorithms without coordination with other clients. We sketch the training process of the VAFL algorithm.

- The **server** maintains the latest embeddings $\{h_i^{(m)} \mid i \in [N], m \in [M]\}$ and waits for a message from an active client $m$. The message contains either an embedding update or a gradient query.

1. **Update**: Client $m$ sends the embeddings $\{\hat{h}_i^{(m)} \mid i \in B_m\}$ to the server. The server then updates its latest embeddings by $h_i^{(m)} = \hat{h}_i^{(m)}$ for all $i \in B_m$.
2. **Query**: Client $m$ requests the partial gradient with respect to batch $B_m$. The server then sends back to the client $m$ the partial gradient as

$$\left\{ g_i := \frac{\partial f(h_i; y_i)}{\partial h_i} \;\middle|\; i \in B_m, h_i = \sum_{m=1}^{M} h_i^{(m)} \right\}. \tag{3}$$

- A **client** $m$ keeps executing the following steps:
  1. Randomly select a batch $B_m \subset [N]$ s.t. $|B_m| = \tau_m$ (Definition 1) and compute local embeddings $\{\hat{h}_i^{(m)} := \langle \theta^{(m)}, x_i^{(m)} \rangle \mid i \in B_m\}$.
  2. Upload embeddings $\{\hat{h}_i^{(m)} \mid i \in B_m\}$ to the server.
  3. Query gradient from the server and update the local model as $\theta_m \leftarrow \theta_m - \frac{\eta_m}{|B_m|} \sum_{i \in B_m} g_i x_i^{(m)}$, where $\eta_m$ is the local learning rate and $g_i$ is defined in Equation 3.

# B PROOFS OF THEORETICAL RESULTS

We first introduce some notions being used. The following special classes of functions (Nagaraj et al., 2019) are considered.

**Definition 4.** *Consider a differentiable function $f : \mathbb{R}^n \to \mathbb{R}$ and any points $x, y \in \mathbb{R}^n$. We have the following definitions.*

- *$f$ is **convex** if $f$ satisfies*
$$f(y) \geq f(x) + \langle \nabla f(x), y - x \rangle;$$
- *$f$ is $\mu$-**strongly convex** for some $\mu > 0$ if $f$ satisfies*
$$f(y) \geq f(x) + \langle \nabla f(x), y - x \rangle + \frac{\mu}{2} \|x - y\|_2^2;$$
- *$f$ is $G$-**Lipschitz** for some $G > 0$ if $f$ satisfies*
$$|f(x) - f(y)| \leq G\|x - y\|_2;$$
- *$f$ is $L$-**smooth** for some $L > 0$ if $f$ satisfies*
$$\|\nabla f(x) - \nabla f(y)\|_2 \leq L\|x - y\|_2.$$

Meanwhile, we uses the notions of $\epsilon$-net and covering number (Udell & Townsend, 2019), which are widely used tools in high-dimensional probability.

**Definition 5.** *Let $K$ be a compact subset of $\mathbb{R}^n$. A subset $N \subseteq K$ is called an $\epsilon$-net for $K$ if, for every $x \in K$, there exists $y \in N$ such that $\|x - y\|_2 \leq \epsilon$.*

*The minimum cardinality of an $\epsilon$-net for $K$ is called the covering number of $K$ and is represented by $\mathcal{N}(K, \epsilon)$.*

## B.1 PROOF OF THEOREM 1: FAIRNESS GUARANTEE

Formally, given a utility function $U$, the corresponding Shapley value should satisfy four fundamental requirements.

- **Symmetry.** For any two clients $i, j \in [M]$, if for any subset of clients $S \subseteq [M] \setminus \{i, j\}$, $U(S \cup \{i\}) = U(S \cup \{j\})$, then $s_i = s_j$.
- **Zero element.** For any client $i \in [M]$, if for any subset of clients $S \subseteq [M] \setminus \{i\}$, $U(S \cup \{i\}) = U(S)$, then $s_i = 0$.
- **Additivity.** If the utility function $U$ can be decomposed into the sum of separate utility functions, i.e., $U = U_1 + U_2$ for some $U_1, U_2 : 2^{[M]} \to \mathbb{R}$, then for any client $i \in [M]$, $s_i = s_i^1 + s_i^2$, where $s^1$ and $s^2$ are the evaluation metrics associated with the utility functions $U_1$ and $U_2$, respectively.
- **Balance.** $U([M]) = \sum_{i \in [M]} s_i$.

It has been showed that the Shapley value, computed by

$$s_m = \frac{1}{M} \sum_{S \subseteq [M] \setminus \{m\}} \frac{1}{\binom{M-1}{|S|}} \left[ U(S \cup \{m\}) - U(S) \right], \tag{4}$$

is the only metric that satisfies the above requirements (Dubey, 1975; Ghorbani & Zou, 2019).

To prove Theorem 1, we only need to show that the VerFedSV (Definition 2) matches the expression of the classical Shapley value, given the utility function specified by Equation 1 and $U(S) = \frac{1}{T} \sum_{t=1}^{T} U_t(S)$.

*Proof.* We notice that the VerFedSV can be expressed as

$$
\begin{aligned}
s_m & \\
&= \frac{1}{MT} \sum_{t=1}^{T} \sum_{S \subseteq [M] \setminus \{m\}} \frac{1}{\binom{M-1}{|S|}} [U_t(S \cup \{m\}) - U_t(S)] \\
&= \frac{1}{M} \sum_{S \subseteq [M] \setminus \{m\}} \frac{1}{\binom{M-1}{|S|}} \left[ \frac{1}{T} \sum_{t=1}^{T} U_t(S \cup \{m\}) - \frac{1}{T} \sum_{t=1}^{T} U_t(S) \right] \\
&= \frac{1}{M} \sum_{S \subseteq [M] \setminus \{m\}} \frac{1}{\binom{M-1}{|S|}} [U(S \cup \{m\}) - U(S)],
\end{aligned}
$$

which matches the expression of the classical Shapley value. The result then follows. $\square$

Another interesting property of VerFedSV is **Periodic additivity.** Formally, If in each iteration, the utility function $U_t$ can be expressed as the sum of separate utility functions, i.e., $U_t = U_t^1 + U_t^2$ for some $U_t^1, U_t^2 : 2^{[M]} \to \mathbb{R}$, then for any client $m \in [M]$, $s_m = s_m^1 + s_m^2$, where $s_m^1$ and $s_m^2$ denotes respectively the VerFedSV computed w.r.t the utility functions $U_t^1$ and $U_t^2$.

### B.2 PROOF OF PROPOSITION 1: BOUNDS OF THE EMBEDDING MATRIX'S $\epsilon$-RANK

*Proof.* It is evident that $\text{rank}_\epsilon(\mathcal{H}^{(m)}) \leq \text{rank}(\mathcal{H}^{(m)}) \leq d^{(m)}$ for all $m \in [M]$. So we only need to prove the remaining two upper bounds. First, we consider the difference between successive rows of the embedding matrix $\mathcal{H}^{(m)}$. For any $t \in [T-1]$ and $i \in [N]$,

$$
\begin{aligned}
|\mathcal{H}_{t,i}^{(m)} - \mathcal{H}_{t+1,i}^{(m)}| &= |\left(h_i^{(m)}\right)^{(t)} - \left(h_i^{(m)}\right)^{(t+1)}| \\
&= |\langle \theta_m^{(t)}, x_i^{(m)} \rangle - \langle \theta_m^{(t+1)}, x_i^{(m)} \rangle| \\
&= |\langle \theta_m^{(t)} - \theta_m^{(t+1)}, x_i^{(m)} \rangle| \\
&= \left| \left\langle \frac{\eta^{(t)}}{|B^{(t)}|} \sum_{j \in B^{(t)}} g_j^{(t)} x_j^{(m)}, x_i^{(m)} \right\rangle \right| \\
&\leq \eta^{(t)} L.
\end{aligned}
$$

Thus, we obtain an upper bound on $\text{rank}_\epsilon(\mathcal{H}^{(m)})$ by

$$
\begin{aligned}
\text{rank}_\epsilon(\mathcal{H}^{(m)}) &\leq \left\lceil \frac{1}{\epsilon} \sum_{t=1}^{T-1} \|\mathcal{H}^{(m)}[t,:] - \mathcal{H}^{(m)}[t+1,:]\|_{\max} \right\rceil \\
&\leq \left\lceil \frac{L}{\epsilon} \sum_{t=1}^{T-1} \eta^{(t)} \right\rceil \\
&\leq \left\lceil \frac{L \log(T)}{\epsilon} \right\rceil.
\end{aligned}
$$

Next, we consider the difference between any two columns of the embedding matrix $\mathcal{H}^{(m)}$. For any $t \in [T]$ and $i, j \in [N]$, we have

$$
\begin{aligned}
|\mathcal{H}_{t,i}^{(m)} - \mathcal{H}_{t,j}^{(m)}| &= |\left(h_i^{(m)}\right)^{(t)} - (h_j^{(m)})^{(t)}| \\
&= |\langle \theta_m^{(t)}, x_i^{(m)} \rangle - \langle \theta_m^{(t)}, x_j^{(m)} \rangle| \\
&\leq \|\theta_m^{(t)}\| \cdot \|x_i^{(m)} - x_j^{(m)}\|.
\end{aligned}
$$

It follows that

$$\|\mathcal{H}^{(m)}[:,i] - \mathcal{H}_{t,j}^{(m)}\|_{\max} \leq \max_{t \in [T]} \|\theta_m^{(t)}\| \cdot \|x_i^{(m)} - x_j^{(m)}\|.$$

Let $\gamma^{(m)} = \max_{t \in [T]} \|\theta_m^{(t)}\|$ and $\mathcal{N}$ be an $\frac{\epsilon}{\gamma^{(m)}}$-net for $\{x_i^{(m)} : i \in [N]\}$. We can thus conclude that $\mathrm{rank}_\epsilon(\mathcal{H}^{(m)}) \leq |\mathcal{N}|$. By definition of the covering number, it follows that

$$\mathrm{rank}_\epsilon(\mathcal{H}^{(m)}) \leq \mathcal{N}\left(\{x_i^{(m)} : i \in [N]\}, \frac{\epsilon}{\gamma^{(m)}}\right).$$

$\square$

### B.3 PROOF OF PROPOSITION 2: APPROXIMATION GUARANTEE

*Proof.* Define $U, \hat{U} : 2^{[M]} \to \mathbb{R}$ by $U(S) := \frac{1}{T}\sum_{t=1}^T U_t(S)$ and $\hat{U}(S) := \frac{1}{T}\sum_{t=1}^T \hat{U}_t(S)$. For any $S \subset [M]$, we know that

$$
\begin{aligned}
&|U(S) - \hat{U}(S)| \\
\leq& \frac{1}{NT}\sum_{t=1}^T\sum_{i=1}^N \left| f\left(\sum_{m=1}^M \mathcal{H}_{t-1,i}^{(m)}; y_i\right) - f\left(\sum_{m=1}^M (w_{t-1}^{(m)})^\intercal h_i^{(m)}; y_i\right) \right| \\
&+ \frac{1}{NT}\sum_{t=1}^T\sum_{i=1}^N \left| f\left(\sum_{m \in S}\mathcal{H}_{t,i}^{(m)} + \sum_{m \notin S}\mathcal{H}_{t-1,i}^{(m)}; y_i\right) \right. \\
&\left. - f\left(\sum_{m \in S}(w_t^{(m)})^\intercal h_i^{(m)} + \sum_{m \notin S}(w_{t-1}^{(m)})^\intercal h_i^{(m)}; y_i\right) \right| \\
\leq& \frac{G}{NT}\sum_{t=1}^T\sum_{i=1}^N \left| \sum_{m=1}^M \mathcal{H}_{t-1,i}^{(m)} - \sum_{m=1}^M (w_{t-1}^{(m)})^\intercal h_i^{(m)} \right| \\
&+ \frac{G}{NT}\sum_{t=1}^T\sum_{i=1}^N \left| \left(\sum_{m \in S}\mathcal{H}_{t,i}^{(m)} + \sum_{m \notin S}\mathcal{H}_{t-1,i}^{(m)}\right) \right. \\
&\left. - \left(\sum_{m \in S}(w_t^{(m)})^\intercal h_i^{(m)} + \sum_{m \notin S}(w_{t-1}^{(m)})^\intercal h_i^{(m)}\right) \right| \leq GM\epsilon.
\end{aligned}
$$

Then we can obtain a bound on $|s_m - \hat{s}_m|$ by

$$
\begin{aligned}
&|s_m - \hat{s}_m| \\
\leq& \frac{1}{M}\sum_{S \subseteq [M]\setminus\{m\}} \frac{1}{\binom{M-1}{|S|}} \left| U(S \cup \{m\}) - \hat{U}(S \cup \{m\}) \right| \\
&+ \left| U(S) - \hat{U}(S) \right| \\
\leq& 2G\epsilon.
\end{aligned}
$$

$\square$

### B.4 PROOF OF PROPOSITION 3: MORE WORK LEADS TO MORE REWARDS

*Proof.* We know that $\theta_1^*$ and $\theta_2^*$ are the optimal variables for the following convex optimization problem $\min_{\theta_1,\theta_2} \frac{1}{N}\sum_{i=1}^N f(\langle \theta_1, x_i\rangle + \langle \theta_2, x_i\rangle; y_i) = g(\theta_1 + \theta_2)$.

Since $\theta^*$ is the unique minimizer for $g(\theta)$, it follows that $\theta_1^*$ and $\theta_2^*$ must satisfy $\theta_1^* + \theta_2^* = \theta^*$. Denote $d_1^{(t)}$ as the update for $\theta_1$ at the $t$-th iteration and $d_2^{(t)}$ as the update for $\theta_2$ at the $t$th iteration. By the construction, we know that

$$\mathbb{E}\left[\sum_{t=1}^\infty (d_1^{(t)} + d_2^{(t)})\right] = \theta^* \text{ and } \mathbb{E}\left[\sum_{t=1}^\infty d_2^{(t)}\right] = \rho\mathbb{E}\left[\sum_{t=1}^\infty d_1^{(t)}\right].$$

Table 2: Metadata of all data sets.

|  | Adult | Web | Covtype | RCV1 |
|---|---|---|---|---|
| number of training data $N$ | 48842 | 119742 | 581012 | 15564 |
| number of features $d$ | 123 | 300 | 54 | 47236 |
| number of classes $l$ | 2 | 2 | 7 | 53 |
| number of clients $M$ | 8 | 20 | 14 | 19 |

Table 3: Hyperparameter setting under synchronous setting.

|  | Adult | Web | Covtype | RCV1 |
|---|---|---|---|---|
| learning rate $\eta$ | 0.2 | 0.2 | 0.8 | 1.0 |
| batch size $\tau$ | 2837 | 6173 | 2000 | 500 |
| rank parameter $r$ | 3 | 3 | 5 | 10 |
| regularization parameter $\lambda$ | 0.1 | 0.1 | 0.1 | 0.1 |

Therefore, it follows that

$$\mathbb{E}[\theta_1^*] = \mathbb{E}\left[\sum_{t=1}^{\infty} d_1^{(t)}\right] = \frac{1}{1+\rho}\theta^* \text{ and}$$

$$\mathbb{E}[\theta_2^*] = \mathbb{E}\left[\sum_{t=1}^{\infty} d_2^{(t)}\right] = \frac{\rho}{1+\rho}\theta^*.$$

Now we consider VerFedSV. By definition, we have

$$\mathbb{E}[s_1 - s_2] = 2\left[g\left(\frac{\rho}{1+\rho}\theta^*\right) - g\left(\frac{1}{1+\rho}\theta^*\right)\right].$$

Define $h : [0,1] \to \mathbb{R}$ as $h(\lambda) = g(\lambda\theta^*)$. Since $g$ is $\mu$-strongly convex and $\theta^*$ is the unique minimizer, it follows that $h$ is $\mu\|\theta^*\|^2$-strongly convex and is monotonically non-increasing on $[0,1]$. Thus, by property of strongly convex (Rockafellar, 1997), we conclude that

$$\mathbb{E}[s_1] \geq \mathbb{E}[s_2] + 2\left[h\left(\frac{\rho}{1+\rho}\right) - h\left(\frac{1}{1+\rho}\right)\right]$$

$$\geq \mathbb{E}[s_2] + \mu\left(\frac{1-\rho}{1+\rho}\right)^2\|\theta^*\|^2.$$

$\square$

## C  DETAILS OF EXPERIMENTAL SETUP

**Data sets**  We use four real-world data sets, *Adult* (Zeng et al., 2008), *Web* (Platt, 1998), *Covtype* (Blackard & Dean, 1999), and *RCV1* (Lewis et al., 2004)[1]. On each data set, by default, we separate the features across *the largest number of clients* reported in the literature (Achen, 1982; Wang et al., 2019; Han et al., 2021; Brown et al., 2012). That is, we compare with the baselines under the most challenging settings. These data sets cover both binary and multiclass classification problems. We summarize the metadata as well as the number of clients for all the datasets in Table 2.

**Models**  For the *Adult*, *Web*, and *Covtype* data sets, we train multinomial logistic regression models, and the local embeddings are computed via a linear model. For the *RCV1* data set, we still use the negative log-likelihood loss, but every client locally has a 2-layer perceptron model with 32 hidden neurons and a ReLU activation function for embedding. Under both the synchronous and asynchronous settings, we can achieve the test accuracy of $85\%$ on the *Adult* dataset, $94\%$ on the *Web* dataset, $72\%$ on the *Covtype* dataset, and $83\%$ on the *RCV1* dataset.

---

[1]From the website of LIBSVM (Chang & Lin, 2011) `https://www.csie.ntu.edu.tw/~cjlin/libsvmtools/datasets/`

Table 4: Hyperparameter setting under asynchronous setting.

|  | Adult | Web | Covtype | RCV1 |
|---|---|---|---|---|
| learning rate $\eta$ | 0.2 | 0.2 | 0.8 | 1.0 |
| batch size $\tau$ | 2837 | 6137 | 2000 | 500 |
| communication frequency $\Delta t$ | 0.01 | 0.01 | 0.01 | 0.01 |

**Monte-Carlo sampling**  In Section D.1, we will describe the adopted sampling method. The VerFedSV is computed exactly when the number of clients $M \leq 10$, and approximately using sampling when the number of clients $M > 10$. The number of randomly sampled permutations $K$ is set as $\lceil 100M \log(M) \rceil$, where $M$ is the number of clients.

**Implementation**  We implement the VFL algorithms and the corresponding VerFedSV computation schemes described in Sections 5 and 6 in the Julia language (Bezanson et al., 2017). The implementations of both synchronous and asynchronous VFL algorithms, as well as the VerFedSV computation schemes, are attached in the supplementary material. The matrix completion problem in Equation 2 is solved by the Julia package `LowRankModels.jl` (Udell et al., 2016). All the experiments are conducted on a Linux server with 32 CPUs and 64 GB memory.

**Hyperparameter setting (synchronous)**  We summarize the hyperparameters under the synchronous setting in Table 3, where $\eta$ and $\tau$ are the learning rate and the batch size used in the FedSGD algorithm, and $r$ and $\lambda$ are the rank parameter and regularization parameter used in the matrix completion problem 2.

**Hyperparameter setting (asynchronous)**  We summarize the hyperparameters under the asynchronous setting in Table 4, where $\eta$ and $\tau$ are the learning rate and the batch size used in the VAFL algorithm, and $\Delta t$ is the communication frequency for clients.

# D  ADDITIONAL EXPERIMENTS AND RESULTS

## D.1  COMPUTATIONAL EFFICIENCY OF VERFEDSV

We first introduce the method to efficiently estimate the VerFedSV. From Definition 2, we can see that the computational complexity for VerFedSV is exponential in the number of clients $M$. The same situation appears in the computation of the classical Shapley value. How to efficiently estimate the Shapley value has been studied extensively (Ghorbani & Zou, 2019; Jia et al., 2019). Many existing methods can be adopted for the computation of VerFedSV. Here we describe the well-known Monte-Carlo sampling method (Metropolis & Ulam, 1949; Ghorbani & Zou, 2019). We can rewrite the definition of VerFedSV into an equivalent formulation using expectation,

$$s_m = \mathop{\mathbb{E}}_{\pi \sim \Pi([M])} \left[ \frac{1}{T} \sum_{t=1}^{T} \Big( U_t(\pi(m) \cup \{m\}) - U_t(\pi(m)) \Big) \right], \qquad (5)$$

where $\Pi([M])$ is the uniform distribution over all $M!$ permutations of the set of clients $[M]$ and $\pi(m)$ is the set of clients preceding client $m$ in permutation $\pi$. With this formulation, we can use the Monte Carlo method to obtain an approximation of VerFedSV. More precisely, we can randomly sample $K$ permutations $\pi_1, \dots, \pi_K$ and approximate VerFedSV $s_m$ by

$$\hat{s}_m = \frac{1}{KT} \sum_{k=1}^{K} \sum_{t=1}^{T} [U_t(\pi_k(m) \cup \{m\}) - U_t(\pi_k(m))]. \qquad (6)$$

By applying Hoeffding's inequality, it can be shown (Jia et al., 2019) that if $K \geq \frac{2R^2 M}{\epsilon} \log\left(\frac{2M}{\delta}\right)$ for some $\epsilon > 0$ and $\delta \in (0,1)$, where

$$R := \max_{S \subseteq [M]} \left[ \frac{1}{T} \sum_{t=1}^{T} U_t(S) \right] - \min_{S \subseteq [M]} \left[ \frac{1}{T} \sum_{t=1}^{T} U_t(S) \right],$$

then we have $\mathbb{P}(|\hat{s}_m - s_m| \leq \epsilon) \geq 1 - \delta$.

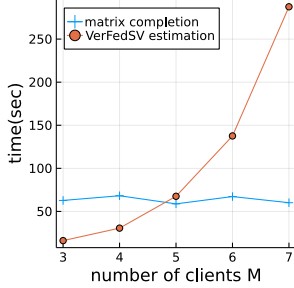 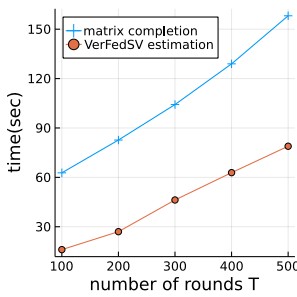 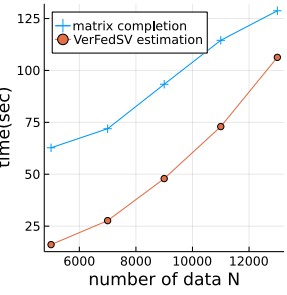

Figure 4: The efficiency of matrix completion and VerFedSV estimation. The left, middle, and right figures show the run time with varying numbers of clients, time-stamps, and data points respectively.

Table 5: Kendall's rank correlation between the results of SHAP and VerFedSV's.

|  | Adult | Web | Covtype | RCV1 |
|---|---|---|---|---|
| $\mathrm{corr}(\mathrm{SHAP}, \mathrm{VerFedSV}_s)$ | 1.0 | 0.73 | 0.94 | 0.65 |
| $\mathrm{corr}(\mathrm{SHAP}, \mathrm{VerFedSV}_a)$ | 1.0 | 0.69 | 0.89 | 0.63 |
| $\mathrm{corr}(\mathrm{VerFedSV}_s, \mathrm{VerFedSV}_a)$ | 1.0 | 0.80 | 0.94 | 0.80 |

In summary, the computation for estimated VerFedSV $\hat{s}_m$ requires $\mathcal{O}(TM\log(M))$ calls of the utility function $U_t$ for both synchronous and asynchronous federated learning algorithms, and the computational complexity for evaluating $U_t$ is $\mathcal{O}(MN)$. Moreover, according to Equation 6, the computation for estimated VerFedSVs can be parallelized. We numerically demonstrate this computational complexity bound in the following.

In this experiment, we test the efficiency of computing VerFedSV. As we can see from Section D.1, the time complexity of estimating VerFedSV is the same for both synchronous and asynchronous VFL algorithms (they only differ in the training time). So we experiment under the synchronous VFL setting. We test the impact of the number of clients $M$, the number of time-stamps $T$ and the number of training data $N$ on time needed for computing VerFedSV. More precisely, we measure the time, respectively, for solving the matrix completion problem and for estimating VerFedSVs. For consistency, the VerFedSV is computed approximately using Monte Carlo regardless of the number of clients in this experiment. Since the time-varying trends are similar for all datasets, we only present the results on the Adult dataset. The result is shown in Figure 4. First, we focus on the time needed for solving the matrix completion problem in Equation 2. As we can see from Figure 4, the runtime keeps stable as the number of clients $M$ changes and scales approximately linearly with the number of time-stamps $T$ and the number of training data $N$, which agrees with our discussion in Section 5. Next, we focus on the time needed for approximating VerFedSV. As illustrated in Figure 4, the runtime exhibits an almost quadratic-logarithmic relationship with the number of clients $M$, and nearly linear relationships with the number of time-stamps $T$ and the volume of training data $N$. This observation aligns with our earlier discussion.

## D.2 EFFECTIVENESS OF VERFEDSV

We evaluate the effectiveness of VerFedSV in both the synchronous and asynchronous VFL settings. In other words, we test whether VerFedSV can reflect the importance of clients' local features. We set all clients to have the same communication frequency to eliminate the impact of unbalanced local computational resources. We choose SHAP (Lundberg & Lee, 2017) as the baseline, which is a widely used metric for measuring feature importance in machine learning. More specifically, we first use SHAP to compute the importance scores of all the features, and then for each client, we ensemble the scores for all the local features. Note that SHAP cannot be directly used in the VFL task, as it requires access to all the local datasets and models and violates the VFL settings. So here we just use it as a reference. We use Kendall's rank correlation (KRC) (Kendall, 1938) to measure the similarity between the orderings of the scores of SHAP and VerFedSV. As a by-product, we also show the similarity between the scores of VerFedSV in the synchronous and asynchronous settings. We show in Table 5 the KRCs between the results of SHAP and VerFedSV, where corr denotes the KRC, SHAP denotes the results from SHAP, VerFedSV$_s$ and VerFedSV$_a$, respectively, denote the results from VerFedSV in the synchronous and asynchronous settings, and the numbers of clients

for the *Adult*, *Web*, *Covtype* and *RCV1* datasets are, respectively, set to $M = 3, 15, 9, 14$. Note that KRC returns a value in $[-1, 1]$, where "1" means two input score lists have the identical ranking and "$-1$" means the rankings are exactly reverted. The KRC between SHAP and VerFedSV are all greater than $0.6$. The results indicate that VerFedSV can indeed capture the feature importance well. Moreover, the KRC between VerFedSV$_s$ and VerFedSV$_a$ are all greater than $0.8$. The results indicate that VerFedSV is consistent under both synchronous and asynchronous settings.

### D.3 ABLATION STUDY ON HYPER-PARAMETERS

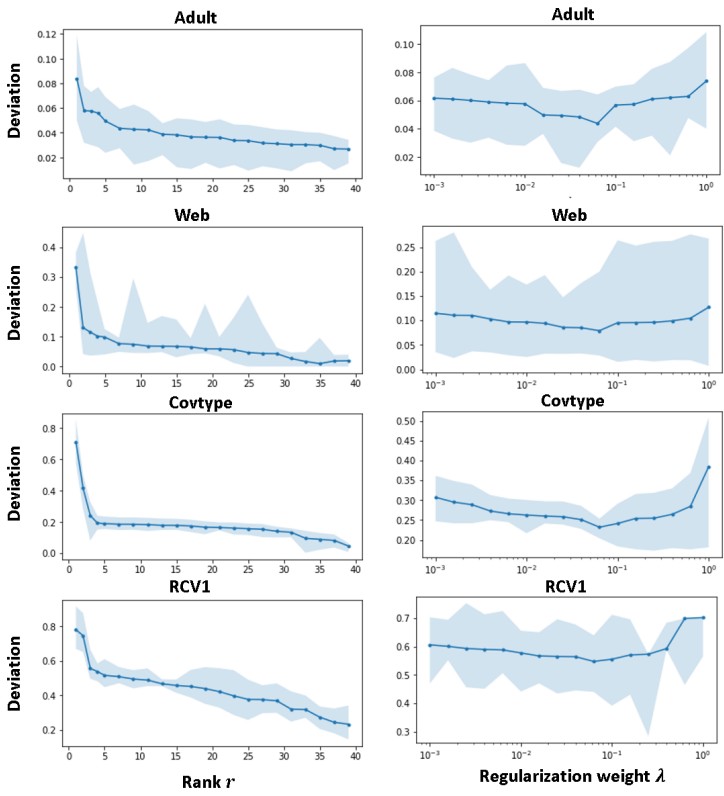

Figure 5: Ablation study on hyper-parameters rank $r$ and regularization $\lambda$

In this section, we examine the sensitivity of VerFedSV with respect to the hyper-parameters $r$ and $\lambda$ used in the matrix completion algorithm, and offer practical guidance for setting these hyper-parameters.

We first recap notations and motivation of matrix completion. As mentioned in Section 5, the key challenge in computing VerFedSV is that, in each iteration, for client $m$, we only have access to a **mini-batch embedding** for the current mini-batch $B$, i.e., $\left\{ h_i^{(m)} \mid i \in B \right\}$. However, computing VerFedSV requires **full embeddings** for **all** training points in each iteration, i.e., $\left\{ h_i^{(m)} \mid i \in [N] \right\}$, where $N$ is the number of training points. Therefore, matrix completion is employed to recover the missing embeddings.

Then, we define a deviation metric for the upcoming experiment. For client $i$, denote its percentage of VerFedSV relative to the total VerFedSV from all clients as $s_i$. The computation process for VerFedSVs involves utilizing matrix completion to estimate full embeddings. Meanwhile, we compute a ground-truth Shapley value, wherein full embeddings are computed in each training round. The percentage of this ground-truth Shapley value for client $i$ relative to the sum is denoted as $s_i^{gt}$. With $M$ representing the number of clients, the deviation metric is defined as $\frac{1}{M} \sum_{m=1}^{M} \frac{|s_m - s_m^{gt}|}{s_m^{gt}}$. This metric necessitates that $s_m^{gt}$ be positive to ensure it is meaningful.

Table 6: The ground-truth percentage of Shapley value relative to the total sum, denoted by $s_i^{gt}$

|          | Adult | Web   | Covtype | RCV1  |
|----------|-------|-------|---------|-------|
| Client 1 | 46.95 | 57.36 | 55.41   | 81.14 |
| Client 2 | 17.43 | 23.21 | 34.54   | 5.10  |
| Client 3 | 35.62 | 19.43 | 10.55   | 13.86 |

For the experiment setup here, the number of clients $M$ is set to 3. Meanwhile, since the clients equally partition the features of a dataset, each client has some contribution to the FL model. Consequently, we observe that the ground-truth Shapley values $s_m^{gt}$ for all clients are positive (See Table 6). The other considerations for setting $M = 3$ include: 1) This experiment does not focus on scalability w.r.t the numbers of clients $M$; 2) We want to avoid Monte Carlo sampling that could potentially interfere with the results. VerFedSV is computed exactly when $M \leq 10$ by our default setup. The other setups remain consistent with those detailed in Table 2 and Table 3. For each pair of hyper-parameters, the ground-truth Shapley value is computed once, while VerFedSV is computed 5 times. In each independent run, each client has the same set of features, but a different random seed, so the matrix completion algorithm may give different estimations of full embeddings.

The left part of Figure 5 illustrates the deviation w.r.t the rank $r$. The median deviation of different runs is depicted by the solid line, while the shaded area represents the interquartile range (25% to 75% quantiles) of deviations. Take the Adult dataset as an example for analysis. The figure demonstrates that deviations are large for $r < 5$, since inadequate rank cannot accurately recover full embeddings. However, deviation decreases rapidly as $r$ increases and stabilizes at a low value when $r \geq 5$.

The default ranks (See Table 3) in the previous experiments were set based on approximated $\epsilon$-rank (See Figure 1). However, in practice obtaining full embeddings is resource-intensive due to communication costs or limited resources on the client side. Thus, approximating $\epsilon$ rank may be difficult. However, the server (with sufficient computational resources), after collecting the mini-batch embeddings, can test with several increasing values of $r$. The Shapley value will converge to the ground-truth value as revealed by our ablation study.

The right part of Figure 5 illustrates the deviation w.r.t the hyper-parameter $\lambda$. It demonstrates that the performance of VerFedSV remains relatively insensitive across a wide range of $\lambda$ and we can safely set $\lambda = 0.1$ for various datasets.

