# OpenReview forum: "Fair and Efficient Contribution Valuation for Vertical Federated Learning"
_ICLR.cc/2024/Conference — ICLR 2024 poster_

### Official Review · Reviewer_WWnM · 2023-10-30

**Soundness:** 3 good
**Presentation:** 3 good
**Contribution:** 2 fair
**Rating:** 6
**Confidence:** 4

**Summary:**

This paper seeks to assign each client a fair value in the vertical federated learning setting where clients share the same sample IDs but own different features. The paper proposes VerFedSV which consider clients’ contributions at multiple time stamps during training. VerFedSV satisfy desirable properties of fairness, can be efficiently computed without model training and extra communication cost.
For the synchronous setting, computing VerFedSV will require the embedding of all data points. However, only the embedding for a mini-batch is available. The paper proposed how to obtain the embedding matrix through low rank matrix completion and show that error in the estimated VerFedSV is bounded.
For the asynchronous setting, VerFedSV can be directly and exactly computed for the mini batch. VerFedSV will reward clients with more computational resources (batch size) and frequent communication.

**Strengths:**

- The paper is well-written. The content and notations are easy to follow.
- The study of related work is comprehensive and it is clear how the paper contrasts with prior work.
- The assumptions (e.g., low rank embedding matrix) are sound and justified (e.g., due to similarity between local data points and during successive training).
- The paper novelly consider data valuation in the _vertical_ federated learning setting and the _asynchronous_ setting with different local computational resources.

**Weaknesses:**

- The paper did not discuss or empirically compare to alternatives, e.g., why not request embedding for all data points or value based on the updated mini-batch or a separate validation set? The experiments lack comparison to a ground-truth (request all embeddings) and other baselines (VFL valuation methods).
- The solution introduced in the work does not seem very novel after considering existing VFL literature and HFL data valuation work. Fan et al. (2022) have used low rank matrix completion to estimate utilities of subsets. In the introduction, it is mentioned that it is more challenging to compute Shapley value to VFL than HFL as it is not applicable (possible) to obtain the global model by concatenating private local models. The challenge has to be further clarified. Is the challenge resolved by using local embedding (as in other VFL work)?

**Questions:**

Questions
1. What are the reasons not to use the asynchronous valuation approach (page 7, use mini-batch only) for the synchronous setting?
2. Existing data valuation works (including those for HFL) usually use a separation validation set for the utility function. If a validation set is used, is the embedding matrix still needed?


Minor suggestions
* The Shapley value is often misspelled as Sharpley, Sharply, Shaprley etc
* The paper can give some examples/explanation of models and $f$ for the reader to understand why the embedding is sufficient. E.g., what is $f$ for linear and logistic regression (in Sec. 3)
* After proposition 1, the notation/arguments for the coverage number should be explained.
* For VAFL, the client has to send the id of the embedding too.
* Report N and T for the experiments

---

> ### Author Response · Authors · 2023-11-22
>
> We sincerely appreciate your positive comments and valuable feedback on our paper. Following your feedback, we have made significant efforts to conduct comparison experiments. We are confident that these efforts have greatly enhanced the quality and strength of our paper. In light of this enhancement to the paper, we kindly hope that you could consider re-evaluating your score. We are confident that our work now better addresses your concerns, and we hope that the updates better reflect the contribution we aim to make to the field. We reply to each of the weaknesses and questions below.
>
> ## Reply to Weakness 1
>
> Thank you for your valuable feedback. We extend the experiments to compare with alternatives including request embeddings for all data points and based on update mini-batch.
>
> We first define a deviation metric for the upcoming comparison experiments. For client $i$, denote its percentage of VerFedSV relative to the total VerFedSV from all clients as $s_i$. These VerFedSVs are calculated using matrix completion to estimate all embeddings. Meanwhile, we compute a ground-truth scenario where full embeddings are calculated in each training round. In this scenario, the percentage of Shapley value for client i relative to the sum of all Shapley values is denoted as $s_i^{gt}$. With $M$ representing the number of clients, the metric Deviation(VerFedSV, Ground truth) is defined as $\frac{1}{M} \sum_{m=1}^{M} \frac{|s_m - s^{gt}_m|}{s^{gt}_m}$.
> Similarly, the metric Deviation(Mini-batch, Ground truth) measures the deviation of Shapley value computed with mini-batch from the ground-truth scenario.
>
> The comparison results are shown as follows. The deviation metric is shown in mean$\pm$std format, where mean and std are calculated from 5 independent runs.
> | | Adult | Web | Covtype | RCV1 |
> |-|-|-|-|-|
> | VerFedSV | $0.058\pm 0.039$| $0.095\pm 0.051$ | $0.309\pm 0.041$ | $0.558\pm 0.117$ |
> | Mini-batch | $0.047\pm 0.150$ | $0.110 \pm0.300$ | $0.319\pm 0.057$ | $0.559\pm 0.268$ |
>
> To conclude, VerFedSV stably achieves low deviation since it can accurately recover the full embeddings. In contrast, the Shapley values under the Mini-batch scenario exhibit a larger variance than VerFedSV.
>
> Preparing a separate validation set under the VFL scenario is a non-trivial research topic. In the VFL scenario, the server should not possess any knowledge about the client dataset, including even the feature dimensions. The server and clients have to collaborate prudently and complicatedly to prepare and utilize the validation set while properly addressing the privacy issue. Considering this challenge, we propose using the full training set directly. This approach offers several advantages. Firstly, it utilizes the entire training set for computing Shapley value, thereby avoiding potential errors introduced by sampling mini-batches. Additionally, it ensures that the computation process of the Shapley value does not leak private information and that the resulting Shapley value remains private to each client.
>
> For baselines, we would like to draw the reviewer's attention to SHAP [1] baseline in Section D.2. This experiment shows that VerFedSV reflects the importance of clients’ local features.
> However, to the best of our knowledge, no other suitable baseline exists that considers both privacy and fairness simultaneously.
> Actually, the SHAP baseline cannot be directly applied to the VFL task, as it requires access to all the local datasets and may breach privacy.
> Meanwhile, the baselines involving the model-independent utility would cause fairness issue in the asynchronous VFL scenario, since they rely on the client's data but ignores the computational resources contributed by the client. To explain this, we have an example in Sec. 7.2. Imagine two clients with identical datasets. Using the model-independent utility function, both clients would be evaluated as having equal contributions. However, in an asynchronous environment, Client 1 may have superior hardware and transmission power. This advantage allows for a larger batch size and facilitates more frequent communication, resulting in a higher number of asynchronous training rounds. In such a scenario, Client 1's contribution should be greater, yet the model-independent utility function fails to recognize this, leading to an unfair assessment.
>
> [2] Scott M. Lundberg, Su-In Lee. A Unified Approach to Interpreting Model Predictions. NeurIPS 2017: 4765-4774

---

> ### Author Response · Authors · 2023-11-22
>
> ## Reply to Weakness 2
>
> Thank you for your feedback regarding the technical novelty of our learning-based basis selection approach. We understand your concerns and appreciate the opportunity to emphasize our contributions:
>
> - We addressed an important problem rarely explored in the literature, that is, how to fairly value the contributions of different clients in the field of vertical federated learning (VFL).
> - Our proposed VerFedSV utilized the local embedding to construct the utility function. We have indeed employed techniques developed by pioneers in related fields (like local embeddings), but our work combines these techniques and applies them in a novel context.
> - VerFedSV is applicable for both synchronous and asynchronous VFL. In synchronous VFL, we utilize matrix completion to obtain the full embedding of all training sets. We demonstrate that the target embedding matrices are approximately low-rank and provide an approximation guarantee for VerFedSV.  Under the asynchronous VFL setting, we show that VerFedSV successfully reflects the strength of clients' local computational resources.
>
> To conclude, our integrated VerFedSV provides a pioneering and exemplary framework for contribution evaluation in VFL, and this approach constitutes a solid contribution.
>
> ## Reply to Question 1
>
> In the asynchronous setting, each client may require different amounts of time to complete a training iteration and may have varying batch sizes. However, in the synchronous setting, all clients have the same time per iteration and the same batch size.
> As described in Section 6, our approach in the asynchronous setting involves the transmission of mini-batches from the clients to the server to update the server embeddings. There is no need to use matrix completion so that the Shapley value is proportional to the client's computational strength.
> In the synchronous setting, we complete the full embedding, because, in this way, the entire training set is utilized for computing Shapley value, effectively avoiding potential errors from mini-batch sampling.
>
> ## Reply to Question 2
>
> Please refer to our response in Weakness 1 regarding preparing and utilizing a separate validation set in the VFL setting.
>
> ## Reply to Minor Suggestions
>
> **Misspellings** We will ensure the misspellings are thoroughly correct in the final version.
>
> **Explanation for why the embeddings are sufficient** For linear regression, $f$ is mean square error, and for logistic regression, $f$ is cross-entropy loss. The client model can be a linear model or a neural network.
> We next explain why VerFedSV only relies on embeddings and is irrelevant to the function form of $f$ or whether the task is one-class or multi-class classification.
>
> We first recap the one-class case and then explain the multi-class case. In one-class case, we defined the embedding matrix $\mathcal{H}^m \in \mathbb{R}^{T \times N}$ where each element $(t,i)$ is defined as $\mathcal{H}_{t,i}^m = (h_i^m)^{(t)}$. In the context of the multi-class classification problem with $l > 2$ classes, the model parameter $\theta_m$ is in space $\mathbb{R}^{d^m \times l}$. Consequently, the embedding $h_i^m = \theta_m^\intercal x_i^m$ is in $\mathbb{R}^l$. Now, the embedding $\mathcal{H}^m$ is a tensor in $\mathbb{R}^{T \times N \times l}$. We reorder the tensor dimensions and lower it to a matrix $\mathcal{H}^m \in \mathbb{R}^{(l\cdot T) \times N}$.
>
> Regarding the general function case, $h_i^m = \theta_m^\intercal x_i^m$ becomes $h_i^m = g(x_i^m; \theta_m)$, where $g$ can be a neural network with weights $\theta_m$. VerFedSV by Definition 2 in Section 4 only requires $h^i_m$ and does not depends on does not depend on the specific form of $g$.
>
> **Explain covering number** Covering number $\mathcal{N}(\mathcal{D},\epsilon)$ represents the minimum number of "balls" or "sets" with radius $\epsilon$ that are needed to cover a given dataset $\mathcal{D}$. In other words, it quantifies the complexity of $\mathcal{D}$. Please refer to Definition 5 in Section B for detailed formulation.
>
> **Clients send the IDs** The client will send the batch $B$ that contains the IDs of data points. We will revise it in the final version.
>
> **Report $N$ and $T$** Please refer to Table 2 in Section C for the number of training data points $N$ in different datasets. As for $T$, under the synchronous setting, it is 160, 160, 800, and 320 for each dataset, and under the asynchronous setting, it is 100. We apologize for any confusion caused and assure to be clarified in the final version.

---

> > ### Comment · Reviewer_WWnM · 2023-11-23
> >
> > Thank you for the detailed response! I have raised my score.

---

> > > ### Author Response · Authors · 2023-11-23
> > >
> > > Thank you for raising the score! Your suggestions have been valuable in helping us improve the quality of our work. We will ensure to incorporate them into the updated pdf.

---

### Official Review · Reviewer_RX3e · 2023-11-01

**Soundness:** 3 good
**Presentation:** 2 fair
**Contribution:** 2 fair
**Rating:** 8
**Confidence:** 3

**Summary:**

This paper proposes a metric for contribution valuation in vertical federated learning (VFL). The method is based on federated Shapley value and is adapted to synchronous VFL via low-rank embedding matrix completion techniques and extended to asynchronous VFL. This metric reflects both the quality of the local dataset and the power of local computational resources.

**Strengths:**

1. The paper is overall well-structured and the authors made good efforts in putting this paper in the literature.

2. Low-rank matrix completion of the embedding matrix resolves the issue that its updates are only partially observed in the mini-batch setting. This technique facilitates the application of Shapley value in VFL.

3. The proposed method VerFedSV is theoretical grounded in this work. It is important to verify that VerFedSV satisfies the criteria of balance, symmetry, zero element, and additivity. Detailed proofs and conditions are provided.

4. The authors experimentally validate the methods on real-world datasets and discussed the results.

**Weaknesses:**

1. Given the setup, it appears that the proposed metric works well only linear models or training only the last linear layer. Although the authors mentioned that results can be extended to a multi-class classification problem and general function, little concrete explanations are provided.

2. There seems be no baselines or comparison with other methods. I understand that contribution evaluation in VFL might not be well-studied before but at least Data Shapley Value can be used here.

**Questions:**

1. In VERFEDSV, would it be inefficient to sum over all subsets $S \subset M$?

2. What is the difference between the method in section 6 and FedSV?

---

> ### Author Response · Authors · 2023-11-22
>
> We sincerely appreciate your positive comments and valuable feedback on our paper. We reply to each of the weaknesses and questions below.
>
> ## Reply to Weakness 1
>
> Thank you for your valuable feedback. Our experiments indeed encompass multi-class (e.g., on Covtype) and general function cases (e.g., 2-layer perceptron model on RCV1). We will ensure providing concrete explanations in the final version.
>
> We first recap the one-class case and then explain the multi-class case. In one-class case, we defined the embedding matrix $\mathcal{H}^m \in \mathbb{R}^{T \times N}$ where each element $(t,i)$ is defined as $\mathcal{H}_{t,i}^m = (h_i^m)^{(t)}$. In the context of the multi-class classification problem with $l > 2$ classes, the model parameter $\theta_m$ is in space $\mathbb{R}^{d^m \times l}$. Consequently, the embedding $h_i^m = \theta_m^\intercal x_i^m$ is in $\mathbb{R}^l$. Now, the embedding $\mathcal{H}^m$ is a tensor in $\mathbb{R}^{T \times N \times l}$. We reorder the tensor dimensions and lower it to a matrix $\mathcal{H}^m \in \mathbb{R}^{(l\cdot T) \times N}$.
>
> Regarding the general function case, $h_i^m = \theta_m^\intercal x_i^m$ becomes $h_i^m = g(x_i^m; \theta_m)$, where $g$ can be a neural network with weights $\theta_m$. Since our method does not depend on the specific form of $g$ and only requires $h_i^m$, it can be extended to the general function case. Admittedly, Proposition 1 on the bound of embedding matrix’s $\epsilon$-rank may not necessarily hold. However, in practice (See Section 7.1, RCV1 dataset) we still observe that embedding is approximately low-rank, so matrix completion is still an effective strategy.
>
> ## Reply to Weakness 2
>
> Thank you for your feedback on the baselines. We would like to mention that data Shapley value [1] can be easily extended to the HFL scenario but not to the VFL scenario. This is because VFL's clients partition the dataset along feature dimensions, while the data Shapley value [1] was designed to assess the importance of individual data points.
>
> We would like to draw the reviewer's attention to our extra experiment in Section D.2, where SHAP [2] is employed as a baseline to test whether VerFedSV reflects the importance of clients’ local features. It is worth noting that the SHAP baseline cannot be directly applied to the VFL task, as it requires access to all the local datasets and may breach privacy. Nevertheless, the results presented in Table 5 of Section D.2 indicate that VerFedSV is indeed capable of capturing feature importance.
>
> [1] Amirata Ghorbani, James Y. Zou. Data Shapley: Equitable Valuation of Data for Machine Learning. ICML 2019: 2242-2251
> [2] Scott M. Lundberg, Su-In Lee. A Unified Approach to Interpreting Model Predictions. NeurIPS 2017: 4765-4774
>
> ## Reply to Question 1
>
> In VerFedSV, it will be inefficient to sum over all subsets $S\subset[M]$. However, this is a common issue in Shapley Value and well-studied in literature. Various sampling methods [1,3] are proposed for this issue. In our experiment, as mentioned in Section C, VerFedSV is computed exactly when the number of clients $M \le 10$, and approximately using Monte-Carlo sampling when the number of clients $M > 10$.
>
> VerFedSV is efficient because it does not require model retraining. The key idea is to calculate the Shapley value for a client in each iteration of training and then aggregate these values over all iterations to determine the final contribution of the client.
>
> [3] Jiayao Zhang, Qiheng Sun, Jinfei Liu, Li Xiong, Jian Pei, Kui Ren. Efficient Sampling Approaches to Shapley Value Approximation. Proc. ACM Manag. Data 1(1): 48:1-48:24 (2023)
>
> ## Reply to Question 2
>
> FedSV [4] and VerFedSV are applied in different scenarios -- HFL and VFL, respectively. Furthermore, it's worth noting that FedSV is specifically designed for the synchronousFL setting and relies on training rounds. In contrast, VerFedSV in Section 6 applies to the asynchronous FL setting.
>
> [4] Tianhao Wang, Johannes Rausch, Ce Zhang, Ruoxi Jia, Dawn Song. A Principled Approach to Data Valuation for Federated Learning. Federated Learning 2020: 153-167

---

> > ### Comment · Reviewer_RX3e · 2023-11-23
> >
> > Thank you for your reply. I have raised my score.

---

### Official Review · Reviewer_n6Bm · 2023-11-04

**Soundness:** 3 good
**Presentation:** 2 fair
**Contribution:** 3 good
**Rating:** 6
**Confidence:** 3

**Summary:**

This paper extends the FedSV framework by Wang et al. (2020) to vertical federated learning setting. For synchronous setting, since the server does not have access to the data embeddings to the full dataset, the authors propose an approximation algorithm based on matrix completion given the low rank assumption. For asynchronous setting, since there are no definition of training iterations for the server, the time stamps for contribution valuation is pre-determined. The author conducted experiments to demonstrate the effectiveness of the proposed technique.

**Strengths:**

The idea of using matrix completion is super interesting.

**Weaknesses:**

The writing can be improved quite a bit.
- For example, $d^m$ is used for denoting the feature dimension of client $m$, but $m$ here can be easily misunderstood as exponent. Better to use $d^{(m)}$.
- Quite a few places use "Sharpley".
- It's good to provide some background about matrix completion algorithm, at least in Appendix.
- In Proposition 1, what is $D^m$? What is $\gamma^m$? Again, $m$ can be easily misunderstood as exponent.
- Proposition 3 can be better stated as an example instead of Proposition.

I am also not too sure about the experiments.
- What is the dimension of each dataset being used?
- What does Table 1 supposed to tell? Is it fair to compare the Shapley value of 1 artificial client against all regular clients?
- How to set the hyperparameters of $r$ and $\lambda$? I think ablation study is needed.

**Questions:**

See weakness.

---

> ### Author Response · Authors · 2023-11-22
>
> We sincerely appreciate the valuable suggestions you provided during the review process. Following your suggestions, we have devoted great effort to an expanded ablation study. We believe that these improvements have strengthened the paper substantially. In light of these efforts and the resulting improvements to the paper, we kindly hope that you could consider re-evaluating your score. We are confident that our work now better addresses your concerns, and we hope that the updates better reflect the contribution we aim to make to the field. We reply to each of the weaknesses and questions below.
>
> ## Reply to Weakness on Writing
>
> Thank you for your suggestions on writing and clarity. We will revise $d^m$ to $d^{(m)}$ to prevent potential misinterpretation and will thoroughly correct the typos.
>
> Concerning the utilization of low-rank matrix completion, we initially introduced its motivation, formulation, and application in Section 5. We will further enrich the backgrounds for its solving algorithm in the appendix, including a taxonomy of existing algorithms and more detailed explanations.
>
> In Proposition 1, $\mathcal{D}^{(m)}$ is the local data set for client $m$ defined in Section 3. Regarding the term $\gamma^{(m)}$, we deeply regret any confusion it may cause. This term is defined in Section B.2 as the bound on the $L_2$ norm of the model parameters. To be specific, it is denoted as $\gamma^{(m)} = \max_{t \in [T]}\|\theta_m^{(t)}\|$.
>
> Finally, in line with your recommendations, we will revise the notation $\cdot^m$ to $\cdot^{(m)}$ and state Proposition 3 as an Example to enhance understanding.
>
> ## Reply to Questions on Experiments
>
> **Question 1** For the dimension of each dataset, please refer to Table 2 (Metadata of all data sets) in the paper.
>
> **Question 2** Table 1 reveals that artificial clients, which submit randomly generated features, are assigned substantially lower Shapley values compared to regular clients providing real features. In this table, the Shapley value of one artificial client is compared against that of the regular clients. This setup’s purpose is to show that artificial clients are indeed evaluated as having lower contributions.
>
> **Question 3** The hyperparameter $r$ is based on approximated $\epsilon$-rank in our experiment. As for $\lambda$, we observe that the performance of VerFedSV remains relatively insensitive across a wide range of $\lambda$. To support these observations, we conduct new ablation studies as follows.
>
> We define a deviation metric for the upcoming ablation study. For client $i$, denote its percentage of VerFedSV relative to the total VerFedSV from all clients as $s_i$. These VerFedSVs are calculated using matrix completion to estimate all embeddings. Meanwhile, we compute a ground-truth scenario where full embeddings are calculated in each training round. In this scenario, the percentage of Shapley value for client i relative to the sum of all Shapley values is denoted as $ s_{i}^{gt} $ . With $M$
> representing the number of clients, the deviation metric is defined as
> $\frac{1}{M} \sum_{m=1}^{M} \frac{|s_m - s^{gt}_m|}{s^{gt}_m} $.
>
> Please refer to this [anonymous link](https://drive.google.com/file/d/14273cwj05-QrdUab_AXa8wSnswo5QHki/view?usp=drive_link) for the results.
>
> The left part of the figure displays the deviation w.r.t the hyperparameter $r$, based on five independent runs. The median deviation of these runs is depicted by the solid line, while the shaded area represents the interquartile range (25% to 75% quantiles) of deviations. Take the Adult dataset as an example for analysis. The figure demonstrates that deviations are significant for $r<5$, since inadequate rank cannot accurately recover full embeddings. However, deviation decreases rapidly as $r$ increases and stabilizes at a low value when $r\ge 5$.
>
> Practically, obtaining full embeddings is resource-intensive due to communication costs or limited resources on the client side. Thus, approximating $\epsilon$ rank may be difficult. However, the server (with sufficient computational resources), after collecting the mini-batch embeddings, can test with several increasing values of $r$. The Shapley value will converge to the ground-truth value as revealed by our ablation study.
>
> The right part of the figure illustrates the deviation w.r.t the hyperparameter $\lambda$. It demonstrates that the performance of VerFedSV remains relatively insensitive across a wide range of $\lambda$ and we can safely set $\lambda=0.1$ for various datasets.

---

> > ### Comment · Reviewer_n6Bm · 2023-11-22
> >
> > Thanks for your reply. For Table 1, I might have some misunderstanding but why not compare it with the **average** Shapley value of regular clients?

---

> > > ### Author Response · Authors · 2023-11-22
> > >
> > > Thank you for your response. We apologize for not addressing the key point in your question. To clarify, in our previous response, we meant to convey that we still conclude that artificial clients are evaluated as having lower contributions, by observing that the artificial client obtains little VerFedSV among the total sum of VerFedSVs (e.g., only 0.01%). However, we admit that it will be better that an artificial client is compared against the average Shapley value of regular clients. We will update the table and ensure to clarify this in the final version.

---

> > > > ### Comment · Reviewer_n6Bm · 2023-11-22
> > > >
> > > > In my opinion, I think a number of 0.01% does not really tell anything to me. There could be billions of regular clients which makes 0.01% a super large number. Furthermore, in data valuation, indirect evaluation metrics such as mislabeled/noisy data detection are also very common (e.g., [1, 2, 3]). I will raise the score if the authors can fix the issue in Table 1.
> > > >
> > > > I like the new ablation study and it's good to incorporate them into the Appendix. One question is that will the groundtruth $g_m^{GT}$ be 0 or negative? Then does the deviation metric make sense in this case?
> > > >
> > > > A side note about related work: Both Data Shapley and distributional Data Shapley are relatively old works (proposed in 2019 and 2020, respectively). I think it's good to review more recent advances in Shapley value-based data valuation (e.g., [1, 2, 3]).
> > > >
> > > > [1] Kwon, Yongchan, and James Zou. "Beta Shapley: a Unified and Noise-reduced Data Valuation Framework for Machine Learning." AISTATS 2022.
> > > >
> > > > [2] Wang, Jiachen T., and Ruoxi Jia. "Data banzhaf: A robust data valuation framework for machine learning." AISTATS 2023.
> > > >
> > > > [3] Wang, Jiachen T., et al. "Threshold KNN-Shapley: A Linear-Time and Privacy-Friendly Approach to Data Valuation." NeurIPS 2023.

---

> > > > > ### Author Response · Authors · 2023-11-23
> > > > >
> > > > > Thank you for your response. We fixed the issue in Table 1. The artificial clients are compared with the average of regular clients now.
> > > > >
> > > > > | | |Adult (Sync) |Web (Sync) | Covtype (Sync) |RCV1 (Sync) | |Adult (Async) |Web  (Async) | Covtype (Async)   |RCV1 (Async) |
> > > > > | --- | --- | -- | --- | --- | --- |--| ---- | ---- | ---- | ---- |
> > > > > |average of regular clients| | 12.48 | 4.98 | 6.98 |4.93| | 12.23 | 4.91 | 7.04 | 4.83 |
> > > > > | artificial client 1 | | 0.01 | 0.02 | 0.39 | 1.86 | | 0.93 | 0.26 | 0.35 | 1.66 |
> > > > > | artificial client 2 | | 0.05 | 0.06 | 0.57 | 0.43 | | 0.06 | 0.31 | 0.21 | 1.75 |
> > > > > | artificial client 3 | | 0.03 | 0.06 | 0.91 | 2.37 | | 0.33 | 0.36 | 0.22 | 1.64 |
> > > > > | artificial client 4 | | 0.01 | 0.07 | 0.36 | 0.31 | | 0.43 | 0.40 | 0.25 | 1.50 |
> > > > > | artificial client 5 | | 0.02 | 0.06 | 0.01 | 1.18 | | 0.35 | 0.28 | 0.39 | 1.54 |
> > > > >
> > > > > The setup of Table 1 is inspired by a similar idea of indirect evaluation (whether VerFedSV can detect clients with noise features). Artificial clients submit random features, and VerFedSV successfully identifies that these clients have a low contribution.
> > > > >
> > > > > In the new ablation study, the $s_i^{gt}$ (which represents ground-truth percentage of Shapley value for client $i$ relative to the sum of all Shapley values) is positive because we set clients equally partition all features and thus each client has some contributions.
> > > > > In other cases, $s_i^{gt}$ can be negative. For example, if a client only has few features, then its marginal contribution may be negative. In these cases, the "percentage of Shapley value" does not make sense, so we first shift all the clients' Shapley values by $|\min_i \{ \text{Client i's Shapley value } \}|$ and then compute the percentage.
> > > > >
> > > > > We appreciate your suggested references and will ensure to review more recent advances, including but not limited to, recent works on Shapley Value and its extension to HFL and VFL.

---

> ### Comment · Reviewer_n6Bm · 2023-11-23
>
> Thanks for your response. I have raised my score. Please also update the pdf.
>
> If all clients' Shapley value is shifted by the minimum of the Shapley value, then is $g_m^{GT}=0$ for the client with the smallest Shapley value?

---

> > ### Author Response · Authors · 2023-11-23
> >
> > Thank you for improving the score! We greatly appreciate your feedback and will ensure to incorporate them into the updated pdf. For the follow-up question, the answer is yes, after shifting, the smallest Shapley value will be 0.

---

> > > ### Comment · Reviewer_n6Bm · 2023-11-23
> > >
> > > If there is a $g_m^{GT}=0$, does the deviation metric still make sense?
> > >
> > > I think it's good for the author to incorporate the committed material into the updated pdf before the end of the discussion period given that the conference provides such an option. It can help the reviewer to understand these additional results in a more comprehensive way and I would then be more comfortable recommending acceptance.

---

> ### Author Response · Authors · 2023-11-23
>
> Thank you for your response. As we mentioned earlier, in the new ablation study (verifying hyper-parameter sensitivity of VerFedSV), $s_i^{gt}$ is positive, because we have this setup: clients equally partition all features and thus each client has some contributions. The deviation metric makes sense in this context.
>
> Considering the above ablation study has already verified the conclusion on hyper-parameter sensitivity, we did not test on more general cases mentioned above (where $s_i^{gt}$ can be negative). We will add the detailed setup, intermediate result (the value of $s_i^{gt}$), and the additional results to the updated pdf now and upload it soon.

---

> > ### Author Response · Authors · 2023-11-23
> >
> > Thank you again for your valuable feedback. We mainly incorporated the material related to the ablation study in the current revision, but we will ensure to incorporate all committed materials in the final pdf.

---

### Official Review · Reviewer_skLQ · 2023-11-07

**Soundness:** 3 good
**Presentation:** 3 good
**Contribution:** 2 fair
**Rating:** 6
**Confidence:** 4

**Summary:**

This paper proposes a new contribution valuation metric called VerFedSV for vertical federated learning. The VerFedSV metric is based on the classic Shapley value, which is a provably fair contribution valuation metric originating from cooperative game theory. The paper shows that VerFedSV satisfies many desirable properties of fairness and is quite adaptable such that it can be applied under both synchronous and asynchronous VFL settings. The paper also provides experimental results that demonstrate the effectiveness of VerFedSV in terms of fairness and efficiency.

**Strengths:**

S1. This paper proposes a new Shapley value-based contribution valuation metric called VerFedSV for both synchronous and asynchronous vertical federated learning.

S2. This paper provides a theoretical analysis of the VerFedSV metric and demonstrates its effectiveness through experimental results.

S3. This paper addresses an important problem in the field of vertical federated learning, namely how to fairly value contributions from
different clients.

**Weaknesses:**

W1. In the last paragraph of Sec. 2, it is said that model-independent utility function itself may cause some fairness issues in asynchronous scenario and cannot fully reflect a client’s contribution. This claim seems not intuitive, it is better to give a concrete example to explain the fairness issues in detail.

W2. There are missing related studies on Shapley Value in federated learning. Please conduct a more detailed literature survey.
[1] Sunqi Heng et al. ShapleyFL: Robust Federated Learning Based on Shapley Value. KDD 2023.
[2] Zhenan Fan et al. Fair and efficient contribution valuation for vertical federated learning. 2022.

W3. In Sec. 3, there aren’t enough explanations on the meaning of embedding h_i^m. How to compute it and how it is used to compute the loss are not mentioned either.

W4. It is said in Sec. 8 that VerFedSV increases when adding identical features, which may indicate the proposed method is fragile and lacking robustness in practical use, especially in asynchronous setting.

W5. There are so many typos. For example, in the first paragraph of Sec. 4, the terminology “Shapley” is mistakenly written as “Sharply”, “Shaprley”, “Sharply” for so many times.

**Questions:**

Beyond the above weak points, I also have the following questions:

Q1. Please explain the fairness issues caused by model-independent utility function in asynchronous scenario in details. It’s better to give an example.

Q2. Is this paper the first work with model-dependent utility functions in VFL? Are there really just two related works in VFL scenario?

Q3. Please give more explanation about the meaning of embedding h_i^m and how to compute it, which is more friendly to read for people not quite familiar with VFL.

---

> ### Author Response · Authors · 2023-11-22
>
> We sincerely appreciate your positive comments and valuable feedback on our paper. We have addressed your comments and made the necessary corrections to improve the paper. We would like to draw your attention to the novelty and broader impact of our research and hope that you might reconsider increasing the score based on the comprehensive merits of our research. Thank you once again for your time and valuable input. We reply to each of the weaknesses and questions below.
>
> ## Reply to Weaknesses
>
> **Regarding Weakness 1:** Thank you for your insightful feedback. The model-independent utility function relies on the client's data but ignores the computational resources contributed by the client. This oversight can lead to unfairness in the asynchronous scenario.
>
> Actually, we have an example in Sec. 7.2. Imagine two clients with identical datasets. Using the model-independent utility function, both clients would be evaluated as having equal contributions. However, in an asynchronous environment, Client 1 may have superior hardware and transmission power. This advantage allows for a larger batch size and facilitates more frequent communication, resulting in a higher number of asynchronous training rounds. In such a scenario, Client 1's contribution should be greater, yet the model-independent utility function fails to recognize this, leading to an unfair assessment. We provided detailed numerical results in Sec. 7.2 to support this example. Based on your valuable feedback, we will enhance the clarity of this explanation in Sec. 2 to better communicate the potential fairness issues arising from the model-independent utility function in asynchronous settings.
>
> **Regarding Weakness 2:** We appreciate your suggested references and acknowledge the importance of a comprehensive literature survey in this area.
> Our work VerFedSV is different from the existing works like ShapleyFL [1]. ShapleyFL applies Shapley Value in FL against malicious clients, while VerFedSV focuses on computing Shapley Value efficiently in the VFL scenario.
> We will include more literature works in the final version of our paper, including but not limited to, the references you suggest, more recent works on Shapley Value and its extension to HFL and VFL.
>
> [1] Sunqi Heng et al. ShapleyFL: Robust Federated Learning Based on Shapley Value. KDD 2023
>
> **Regarding Weakness 3:** Thank you for pointing out the need for a clearer explanation of the embedding $h_i^m$ in Sec. 3. The term $h_i^m$ represents the embedding of a local data point $x_i^m$ from the $m$-th client, transformed by the local model's parameters $\theta_m$. For linear and logistic regression models, this embedding is computed as $h_i^m = \langle \theta_m, x_i^m \rangle = \theta_m^\intercal x_i^m$. Then, each client uploads local embeddings $h_i^m$ to the server. The server aggregates these embeddings to obtain a global embedding $h_i$ for $i$-th data point, i.e., $h_i = \sum_{m=1}^M h_i^m$.
> This global embedding is employed to compute the loss, denoted by $f(h_i,y_i)$. Here, $f$ is a differentiable function. For linear regression, mean square error is employed, and for logistic regression, cross-entropy loss is employed.
>
> We illustrated these concepts in Sec.3 and we will make sure they are more clearly explained in the final version.
>
> **Regarding Weakness 4:** We have included this particular weakness in the future work section. However, it does not mean the weakness cannot be solved. One potential solution is to implement secure statistical testing before FL training and exclude clients with extremely similar datasets. This would be an orthogonal and promising future direction. We will further clarify it in the paper.
>
> **Regarding Weakness 5:**  We sincerely apologize for the typos and will thoroughly correct them.
>
> ## Reply to Questions
>
> **For Question 1**, see our reply to Weakness 1.
>
> **For Question 2**, to our best knowledge, this paper is the first work of model-dependent utility function in VFL. Regarding related works of Shapley value in the VFL, we acknowledge the rapidly evolving dynamics within this research community, so we will surely update the final version of our paper with a comprehensive review.
>
> **For Question 3**, see our reply to Weakness 3.

---

> > ### Author Response · Authors · 2023-11-23
> >
> > Dear Reviewer,
> >
> > I hope this message finds you well. Thank you for taking the time to review our paper. We are looking forward to receiving your follow-up feedback and suggestions. If our response has addressed your concerns, we would highly appreciate it if you could consider raising the score. If you have any additional questions or suggestions, we would be happy to have further discussions. Your feedback is important to us, and we want to ensure that we have addressed all of your comments.
> >
> > Thank you again for your time and consideration.
> >
> > Best regards,
> >
> > Authors

---

> > > ### Comment · Reviewer_skLQ · 2023-11-23
> > > **Response to author feedback**
> > >
> > > I have carefully read the response and I have no further questions.

---

> > > ### Comment · Reviewer_skLQ · 2023-11-23
> > >
> > > I have raised my score.

---

### Meta-Review · Area_Chair_TXuT · 2023-12-08

**Metareview:**

This paper proposes a new provably fair metric called Vertical Federated Shapley Value (VerFedSV) that is efficient to compute for vertical federated learning.

**STRENGTHS**

(1) A novel Shapley value-based contribution valuation metric has been proposed for both synchronous and **asynchronous vertical** federated learning.

(2) This work provides both a theoretical analysis of VerFedSV and empirical evaluation of its effectiveness.

(3) This paper addresses an important problem.


**WEAKNESSES**

(1) This paper has several writing issues/typos, clarity issues, and missing empirical results. The authors have carefully addressed them in the rebuttal.

(2) As pointed out by a number of reviewers, the authors have missed out important references on Shapley values in federated learning as well as collaborative machine learning. In addition to the references provided by the reviewers, I have included others below (see references therein). The authors should provide a discussion on how their work differs or improves upon such existing works:

Fair yet Asymptotically Equal Collaborative Learning. ICML 2023.

Gradient Driven Rewards to Guarantee Fairness in Collaborative Machine Learning. NeurIPS 2021.

Data Valuation in Machine Learning: "Ingredients", Strategies, and Open Challenges. IJCAI 2022.

https://github.com/daviddao/awesome-data-valuation

The authors are strongly encouraged to revise their paper to address the above remaining concerns and the reviewers'.

**Justification For Why Not Higher Score:**

This work extends the use of low-rank matrix completion in horizontal FL (Fan et al. 2022) to that of vertical FL.

To the best of my knowledge, a revised paper has not been submitted even though much changes need to be incorporated, as evident in the rebuttal.

**Justification For Why Not Lower Score:**

The reviewers deem this work to be of sufficient technical merits.

Note, however, that this paper has all the reviewers responding actively to clarify and help in improving the paper, which is a luxury not shared by other submitted papers.

---

### Decision · Program_Chairs · 2024-01-16

Accept (poster)